# Dysbiosis by Eradication of *Helicobacter pylori* Infection Associated with Follicular Gastropathy and Pangastropathy

**DOI:** 10.3390/microorganisms11112748

**Published:** 2023-11-10

**Authors:** Uriel Gomez-Ramirez, Carolina G. Nolasco-Romero, Araceli Contreras-Rodríguez, Gerardo Zuñiga, Sandra Mendoza-Elizalde, Francisco-Javier Prado-Galbarro, Fernando Pérez Aguilar, Jonatan Elihu Pedraza Tinoco, Pedro Valencia-Mayoral, Norma Velázquez-Guadarrama

**Affiliations:** 1Laboratorio de Investigación en Enfermedades Infecciosas, Hospital Infantil de México Federico Gómez, Mexico City 06720, Mexico; urielgoramirez93@outlook.es (U.G.-R.); carolinagnolascor@gmail.com (C.G.N.-R.); hipsme@hotmail.com (S.M.-E.); 2Posgrado en Ciencias Quimicobiológicas, Escuela Nacional de Ciencias Biológicas, Instituto Politécnico Nacional, Mexico City 11340, Mexico; 3Departamento de Microbiología, Escuela Nacional de Ciencias Biológicas, Instituto Politécnico Nacional, Mexico City 11340, Mexico; aracelicontreras21@gmail.com; 4Laboratorio de Variación Biológica y Evolución, Departamento de Zoología, Escuela Nacional de Ciencias Biológicas, Instituto Politécnico Nacional, Mexico City 11340, Mexico; capotezu@hotmail.com; 5Dirección de Investigación, Hospital Infantil de México Federico Gómez, Mexico City 06720, Mexico; frjavipg@gmail.com; 6Servicio de Endoscopía Gastrointestinal, Hospital General Dr. Fernando Quiroz, Instituto de Seguridad y Servicios Sociales de los Trabajadores del Estado, Mexico City 01140, Mexico; ferperezaguilar@gmail.com; 7Clinica Medizama Tulum, Tulum 77760, Mexico; 8Departamento de Patología Clínica y Experimental, Hospital Infantil de México Federico Gómez, Mexico City 06720, Mexico

**Keywords:** *Cutibacterium acnes*, *Helicobacter pylori*, pathobiont, metagenomics, histopathological evaluation, quantitative real-time polymerase chain reaction, bioinformatics

## Abstract

Dysbiosis plays an important role in the development of bacterial infections in the gastric mucosa, particularly *Helicobacter pylori*. The international guidelines for the treatment of *H. pylori* infections suggest standard triple therapy (STT). Nevertheless, because of the increasing resistance rates to clarithromycin, metronidazole has been widely considered in several countries. Unfortunately, the non-justified administration of antibiotics induces dysbiosis in the target organ. We characterized the gastric microbiota of patients diagnosed with follicular gastropathy and pangastropathy attributed to *H. pylori* infection, before and after the administration of STT with metronidazole. Dominant relative abundances of *Cutibacterium* were observed in pre-treatment patients, whereas *H. pylori* was observed at <11%, suggesting the multifactor property of the disease. The correlation of *Cutibacterium acnes* and *H. pylori* with gastric infectious diseases was also evaluated using quantitative real-time polymerase chain reaction. The dominance of *C. acnes* over *H. pylori* was observed in gastritis, gastropathies, and non-significant histological alterations. None of the microorganisms were detected in the intestinal metaplasia. Post-treatment alterations revealed an increase in the relative abundances of *Staphylococcus*, *Pseudomonas*, and *Klebsiella*. Non-*H. pylori* gastrointestinal bacteria can be associated with the initiation and development of gastric diseases, such as pathobiont *C. acnes*.

## 1. Introduction

Gastropathy is a condition that affects the gastric mucosa [1]. It is characterized by the prevalence of mononuclear infiltrates, the formation of lymphoid follicles with germinal centres in the lamina propria, and subsequent tissue inflammation and regeneration. Histological injury is frequently located in the gastric antrum; however, it sometimes extends to the gastric corpus [2]. Gastropathies are classified according to their aetiologies. Follicular gastropathy (also known as reactive or chemical gastropathy) is mainly attributed to irritants, and their recurrent abuse, including alcohol and nonsteroidal anti-inflammatory drugs (NSAIDs) [1], which play an important role in the development of bacterial infections in the mucosa, particularly those induced by *Helicobacter pylori* [3]. Approximately 50% of the worldwide population is colonized with *H. pylori*, representing up to 90% of the gastric microbiota of healthy subjects [4]. However, only 1–2% of this population develops several degrees of chronic mucosal inflammation with clinical manifestations and severe complications [5]. The Mexican consensus on the diagnosis, prevention, and treatment of NSAID-induced gastropathy and enteropathy [6] has determined that the consumption of NSAIDs in subjects with *H. pylori* infection is a risk factor (relative risk = 20.8) for the development of gastroduodenopathies [7]. 

*H. pylori* is classified as a high-priority pathogen because of its resistance to multiple antibiotics [8]. Its association with several infectious diseases has led us to seriously consider the most accurate diagnoses for the optimal selection of treatment therapies. Several non-invasive diagnostic techniques, such as the urea breath test, blood and stool tests, and upper gastrointestinal series [9], have been widely developed for the detection of this bacterium. However, endoscopic studies, which are considered invasive methods for clinical diagnosis, remain helpful because they facilitate the early identification of the aetiological agent [10].

*H. pylori* eradication has been suggested in patients with gastropathy, who are undergoing NSAIDs therapy, to avoid subsequent complications (i.e., development of ulcerations) [6]. International guidelines for the treatment of *H. pylori* infections suggest the use of the standard triple therapy (STT), which is considered the first therapeutic alternative for effective eradication [11]. This therapy consists of two antibiotics and a proton pump inhibitor (PPI), usually composed of clarithromycin (CLR), amoxicillin (AMX), and omeprazole [12]. However, clinical guides for the eradication of *H. pylori* in Mexico and several European countries have replaced the use of CLR by considering metronidazole (MTZ) as a first-line drug for treating this infection [13,14].

Antibiotics have improved the outcomes of treatment therapies for bacterial infectious diseases. They are considered one of the major contributors to the rise in life expectancy by exponentially diminishing morbidity and mortality rates. Unfortunately, the sometimes-unjustified employment and abuse of drugs for the relief of gastric symptomatology or self-treatment of several infectious diseases have contributed to the global crisis of resistance to multiple antibiotics. Although antibiotics are widely employed for the treatment of specific bacterial infections [15], they can cause several states of dysbiosis. 

Dysbiosis, defined as the induction of several alterations in the native microbiota of the host, can be observed as a significant reduction or permanent loss of commensal species in the affected microbial communities and/or the establishment, colonization, or overgrowth of gastrointestinal pathobionts (native commensal microbiota that presents pathogen behaviours under dysbiotic conditions) [16]. Although *H. pylori* induces dysbiosis in infected patients by colonizing up to 90% of the total gastric microenvironment for its establishment, recent studies have highlighted the possible association of non-*H. pylori* bacteria with the initiation and development of not only follicular gastropathy and pangastropathy but also several gastric infectious diseases [17] by correlating bacterial overgrowth under non-favourable microenvironmental conditions [18], such as the pathobiont *Cutibacterium acnes*.

The species of the genus *Cutibacterium* reside as dominant commensal bacteria in the human skin [19]. Although most *Cutibacterium* species are adapted to inhabit the human skin [20,21], *C. acnes* is mainly associated with the maintenance of skin homeostasis due to its multiple benefits in this organ. This species was first reported in healthy gastric mucosa [22,23,24]; however, its functions in the gastric microenvironment have not been fully elucidated. *C. acnes* has recently been classified as a possible trigger of gastric clinical outcomes, such as corpus-dominant lymphocytic gastritis [25], and has been considered a high-risk factor for the initiation and development of gastric cancer [26].

This study aimed to characterize the gastric microbiota of patients with follicular gastropathy and pangastropathy, evaluate the effects of STT on the gastric environment, and correlate the presence of pathobiont *C. acnes* with the development of gastric infectious diseases.

## 2. Materials and Methods

### 2.1. Characterization of the Gastric Microenvironment

#### 2.1.1. Study Subjects and Sample Collection

From June to August 2015, 17 patients presented to the Gastroenterology Service at the Hospital General Dr. Fernando Quiroz Gutiérrez, Instituto de Seguridad y Servicios Sociales de los Trabajadores del Estado (ISSSTE), Mexico City, Mexico, and provided informed consent for gastric sampling. This study was conducted according to the Declaration of Helsinki and was approved by the Institutional Ethics Committee of the Hospital Infantil de México Federico Gómez (protocol code HIM/2017/010 SSA 1305). Patients were selected according to the following inclusion criteria: patients aged 18–65 years, and those who were endoscopically diagnosed with follicular gastropathy and pangastropathy. The exclusion criteria were as follows: patients who did not wish to participate in this study; those who did not follow the protocol indications; and those with medical conditions that could not allow them to stand without assistance. According to the inclusion and exclusion criteria, only six patients were finally included in the study for a second gastric biopsy sampling (Figure 1).

Eighteen biopsies (three pre-treatment samples per patient) were first obtained from the endoscopic study. Six samples were processed for histopathological evaluation. Six additional samples were stored in Brucella broth for microbiological and molecular analyses. The remaining biopsy samples were stored in 1 mL GeneAll^®^ RiboEx for next-generation sequencing (NGS).

According to the symptomatology and the endoscopic study, all patients were diagnosed with follicular gastropathy and follicular pangastropathy associated with *H. pylori* infection. The study subjects were treated with an STT (1 g AMX twice a day, 1 g MTZ per day, and 40 mg omeprazole twice a day) administered for 14 days. The remaining 18 gastric biopsy samples were collected 30 days after the administration of STT (post-treatment samples) and stored as described above.

#### 2.1.2. Histopathological Evaluation

Twelve gastric biopsy samples were paraffined, cut into 10 μm-thick sections, and stained with Haematoxylin-Eosin (H&E), Giemsa, and immunohistochemical staining techniques. Sections were observed and analysed as a double-blind trial. The categorization of histological damage was established using the Updated Sydney grading system [27]. The classification of histological injury was determined by scores (0: absent; 1: minimum; 2: moderate; 3: severe). Additional histological alterations associated with *H. pylori* infection were evaluated as well.

#### 2.1.3. Isolation and Characterization of *H. pylori* Cultures

Twelve gastric biopsies were processed for the isolation of *H. pylori* as follows: samples were placed in frosted glass mortars with 100 μL Brucella broth for maceration. Approximately 100 μL from each macerated tissue were inoculated in Casman solid medium with 7.5% horse defibrinated blood and incubated at 37 °C for up to 5 days under microaerophilic conditions. *H. pylori* primary cultures were subcultured in Casman solid medium using the cross-streak method and incubated at 37 °C for up to 5 days under microaerophilic conditions. Subcultures were identified by Gram staining, catalase, and oxidase to determine their susceptibility profile and genotype.

##### Antibiotic Susceptibility Profiling and Molecular Genotyping

Both primary and secondary cultures were evaluated to determine their susceptibility profiles to AMX, CLR, MTZ, and levofloxacin (LVX) using the minimum inhibitory concentration (MIC) method via the agar dilution test, according to the Clinical Laboratory Standard Institute (CLSI) guidelines. The *Helicobacter pylori* ATCC (American Type Culture Collection, Manassas, VA, USA) 43504 reference strain was used to validate the technique. The breakpoint values (μg/mL) used for the determination of the MIC were taken from CLSI, 2015 [28]. Strains were classified as sensitive (S), intermediate (I), or resistant I.

The subcultures were then processed for DNA isolation using the Wizard Genomic DNA Purification Kit (Promega, Madison, WI, USA), according to the manufacturer’s instructions. The DNA was used to determine the genotypes of each subculture, which was performed via end-point PCR, according to the conditions described by Mendoza-Elizalde et al. [29]. The PCR products were then loaded onto 1.5% agarose gel with SYBR Safe DNA Gel Stain (Thermo-Fisher Scientific, Waltham, MA, USA). The running conditions for electrophoresis were as follows: 90 V, 350 mA per 50 min. Amplicons were visualized in the iBright CL1000 imaging system (Thermo-Fisher Scientific, Waltham, MA, USA).

#### 2.1.4. NGS Sequencing

Twelve gastric biopsy samples were processed for the isolation and purification of total RNA using the GeneAll^®^ RiboEx reagent, and complementary DNA (cDNA) was synthesized using the Jena Biosciences SCRIPT cDNA Synthesis Kit. The samples were shipped to Zymo Research (Irvine, CA, USA) for Illumina MiSeq NGS service by sequencing the 16S rRNA V3V4 hypervariable regions.

##### Metagenomic Analysis: Sequencing Data Quality Control and Processing

The sequences were analysed using Quantitative Insights Into Microbial Ecology (QIIME2) v.2022.4 pipeline [30]. Demultiplexed sequences were processed for denoising, dereplication, clustering, and remotion of chimeras using the DADA2 algorithm, which is included in the QIIME2 pipeline. The retained reads were processed for alignment; taxonomic assignation, performed at 97% sequence identity with the SILVA v132 database [31]; and clustering into phylum and genus operational taxonomy units (OTUs). Representative sequences of dominant OTUs were extracted for multiple alignments using the online Basic Local Alignment Search Tool (BLAST) [32] for species identification. Estimation of Alpha- (Shannon’s (H); Faith’s PD; Simpson’s diversity index; Chao1) and Beta-diversity indexes for pre-treatment and post-treatment patients were also estimated using QIIME2. The sampling depth for the estimation of alpha- and beta-diversity indexes was determined using rarefaction tests. Additionally, Good’s coverage index was also estimated using the QIIME2 pipeline to determine the sampling quality in each sample. Beta-diversity analyses were performed via Principal Coordinate Analysis (PCoA) using a Bray–Curtis dissimilarity distance matrix. Student’s *t*-test, Mann–Whitney *U* test, and analysis of similarity (ANOSIM) analyses were performed as well. The R console v 4.2.1 was used for data visualization [33].

### 2.2. Correlation of the Presence of C. acnes and H. pylori with Gastric Diseases

#### 2.2.1. Study Subjects and Sample Collection

Forty formalin-fixed paraffin-embedded (FFPE) gastric biopsy samples from patients who attended the Pathology Service and who previously signed an informed consent for gastric sampling were obtained and stored at room temperature. The study was also conducted according to the Declaration of Helsinki and was approved by the Institutional Ethics Committee of the Hospital Infantil de México Federico Gómez (protocol code HIM/2017/010 SSA 1305). Patients were selected according to the following inclusion criteria: patients possibly referred for *H. pylori* infection, with non-significant histological alterations, gastritis, gastropathies, or intestinal metaplasia. Two sections (10 μm each) from each FFPE sample were deparaffinized for RNA isolation.

#### 2.2.2. FFPE Deparaffinization for RNA Isolation

FFPE tissues from patients with non-significant alterations (*n* = 5), gastritis (*n* = 17), gastropathies (*n* = 7), and intestinal metaplasia (*n* = 11) (Appendix A) were processed as follows: two sections from each sample were added to 1 mL Xylene and incubated for 3 min at 50 °C. Then, tubes were centrifuged at 13,000 rpm for 2 min to discard the supernatant. This process was repeated until the gastric tissues were paraffin-free. One millilitre of absolute ethanol was then added to each sample. The mixture was incubated at room temperature for 3 min. Tubes were then centrifuged as mentioned above to discard the supernatant. Then, each sample was added to 1 mL of absolute ethanol. Tubes were incubated and centrifuged as described above to discard the supernatant. The tubes were incubated at 50 °C for complete drying of the tissue. After total drying, each sample was rehydrated with 85 μL H_2_O DEPC, and treated with 15 μL of proteinase K (20 mg/mL) and 600 μL of a mixture containing 600 μL Nuclei Lysis Solution (Promega), and 120 μL of ethylenediaminetetraacetic acid (EDTA) 0.5 M at pH 8. Samples were homogenised and incubated at 55 °C overnight. After total digestion of the tissue, each sample was collected in 800 μL TRIzol ^®^ reagent for RNA isolation, according to the manufacturer’s instructions. 

Total RNA from each sample was subjected to double-stranded DNA (dsDNA) remotion and cDNA synthesis using DNAse I (Thermo-Fisher Scientific, Waltham, MA, USA) and the SCRIPT cDNA Synthesis (Jena Bioscience, Jena, Germany) kits, respectively, according to the manufacturer’s instructions.

#### 2.2.3. Quantitative Real-Time PCR (qRT-PCR)

Identification and quantification of *C. acnes* and *H. pylori* in the FFPE samples were performed using qRT-PCR, with the primers for identification of *C. acnes* [34] and *H. pylori* [35], which are enlisted as follows: *C. acnes* 16S rRNA-F: 5′ GCG TGA GTG ACG GTA ATG GGT A 3′; 16S rRNA-R: 5′ TTC CGA CGC GAT CAA CCA 3′; *H. pylori* 16S rRNA-F: 5′ TCG GAA TCA CTG GGC GTA A 3′; and 16S rRNA-R: 5′ TTC TAT GGT TAA GCC ATA GGA TTT CAC 3′.

Two internal standard curves were built for the validation of the qRT-PCR reactions: for *C. acnes*, reactions were performed in 14 μL containing 7 μL qPCR SybrMaster HighROX (Jena Bioscience), 2 pmol forward and reverse primers, and 1:2 serial dilutions of cDNA from a clinical strain of *C. acnes*. For *H. pylori*, reactions were also performed in 14 μL containing 7 μL qPCR SybrMaster HighROX, 4 pmol forward and reverse primers, and 1:10 serial dilutions of cDNA from the *H. pylori* ATCC 26695 reference strain. All reactions were tested in triplicate in the AriaMx Real-Time PCR System (Agilent Technologies, Santa Clara, CA, USA) under the following conditions: initial denaturation at 95 °C for 2 min, followed by 35 cycles of denaturation at 95 °C for 15 s, annealing at 60 °C for 20 s, and fluorescence detection at 72 °C for 10 s. Melting curves were built under the following conditions: denaturation at 95 °C for 30 s, followed by annealing at 60 °C for 20 s, and a final extension at 72 °C for 10 s.

#### 2.2.4. Validation of qRT-PCR, Data Treatment and Statistical Analysis

The Cq values were plotted against the logarithm of the concentrations used for the internal standard curves to obtain the slope of each gene. The efficiency (*E*) of each reaction was calculated using the following formula: *E* = 10^(−1/*slope*)^. For the determination of the number of molecules (*N*) in each gastric sample, the formula *N* = *E^Cq^* was used. All statistical analyses were performed using the Statistical Package for the Social Sciences (SPSS) program (IBM Corp., Armonk, NY, USA).

## 3. Results

### 3.1. Characterization of the Gastric Microenvironment

#### 3.1.1. Histopathological Evaluation

Six patients were diagnosed with follicular gastropathy and pangastropathy, associated with *H. pylori* infection. Table 1 shows the histopathological evaluation performed before and after the administration of STT. 

Mucosal atrophy was minimal in patients 1B and 6B, whereas it was moderate in patient 4B. Patients 2B, 3B, and 5B did not present mucosal atrophy. Cellular infiltrates were also identified. Mononuclear infiltrates were observed, as shown in Figure 2. Patients 3B and 6B presented with minimal mononuclear infiltrates, while 1B and 5B presented moderate activity, and 4B presented with severe activity. No patient presented neutrophilic infiltrates, except for 4B (severe).

Mononuclear infiltrates were minimal in all post-treatment patients, except for patient 4A (moderate) and 5A (severe). Metaplastic tissue was not observed in pre-treatment patients. Moreover, moderate and severe nodulations were identified in patients 3B and 4B, respectively. Foveolar hyperplasic tissue was identified in patients 1B and 6B, whereas patient 4B presented with hyperplasic tissue in up to 50% of the tissue (Figure 3).

Almost all pre-treatment patients (except for patients 1B and 4B) lacked regenerative tissue. The bacterial structures were observed in patients 3B, 4B, and 2A. However, in patient 2, no histological injury was observed.

#### 3.1.2. Molecular Identification of *H. pylori* and Urease Detection

Patients 1B, 5B, and 6B tested positive for *H. pylori* by PCR and culture (Table 2). However, no bacillary structures were found during the histopathological evaluation. Patients 1B and 5B presented a moderate level of mononuclear infiltrates, whereas patient 6B had a minimal level of mononuclear infiltrates. The three remaining patients were considered negative. Moreover, patients 3B and 4B presented with histological injuries and bacillary structures. Urease activity was observed in patients 1B, 5B, and 4B. However, in patient 6B, no such activity was detected. 

#### 3.1.3. Antibiotic Susceptibility Profiling and Molecular Genotyping

The susceptibility profiles of the primary cultures revealed resistance to CLR. Additionally, two isolates were resistant to AMX and LVX, and highly resistant to MTZ (Appendix A). However, the subcultures presented multiple susceptibility profiles (Appendix A).

Molecular genotyping of the subcultures revealed relevant genotypic diversity. The *cagA*+ genotype was dominant. The *vacA s*1 region was the most frequent. High polymorphisms were observed in the *m*-region. The most common genotypic profiles were *cagA*+ *vacA s*1+*m*1+ and *cagA*+ *vacA s*1+*m*1+*m*2+ (Appendix A).

#### 3.1.4. Metagenomic Analysis

From the 362,231 raw reads, 78,053 sequences were finally retained for metagenomic analysis. After quality control, a minimum and maximum frequency of sequences per sample ranging from 4167 to 9811, respectively, was observed in the filtered libraries, with an average mean frequency of 6504.42 sequences per patient. All the samples presented a coverage index value of 1 (Table 3; Appendix A). In total, 440 OTUs were observed. Patient 1 presented with the lowest number of OTUs. Patient 5 presented with the highest pre-treatment and post-treatment OTU values.

The most abundant phyla in pre-treatment patients were Actinobacteria (up to 75.99%), Proteobacteria (up to 46.17%), Chloroflexi (up to 15.83%), and Firmicutes (up to 14.41%). In contrast, Acidobacteriota was absent in all pre-treatment samples. The phylum Bacteroidota almost presented the same behaviour, with relative abundances < 6% in one pre-treatment patient. However, the post-treatment analysis revealed the absence of the phylum Campylobacterota in patient 6B; a decrease in Actinobacteria and Firmicutes, and an increase in Proteobacteria (up to 59.19%), Cyanobacteria (up to 42.86%), and Bacteroidota (up to 10.86%) (Appendix A). 

At the genus level (Figure 4), patient 6B presented with relative abundances of *Helicobacter* < 9%, whereas relative abundances < 1% of this bacterium were determined in patients 3B and 5B; however, it was absent in patient 1B (*H. pylori*-positive patient previously confirmed through microbiological and molecular analyses). *Cutibacterium* was dominant in almost all pre-treatment patients, with relative abundances of up to 36.44%, except for in patient 6B. Non-dominant bacteria were determined in low relative abundances before the administration of STT, e.g., *Nocardioides*, *Kocuria*, *Bifidobacterium*, *Brevundimonas*, *Tetragenococcus*, and *Delftia*. Specific genera involved in the maintenance of homeostasis were observed to increase their relative abundances after the administration of STT, e.g., *Staphylococcus*, *Neisseria*, and *Methylobacterium*. *Cutibacterium* and *Helicobacter* representative sequences were all extracted from the QIIME2 representative sequence files and imported into the BLAST program for taxonomic identification at the species level. All sequences were identified as the species *Cutibacterium acnes* and *Helicobacter pylori* (Appendix A).

Pathobionts of clinical interest increased their relative abundances after the administration of STT, including *Pseudomonas*, *Klebsiella*, *Streptococcus*, *Pantoea*, *Escherichia*, and *Acinetobacter*. *Helicobacter* was present in patient 2A with relative abundances of <1%.

Dominance, diversity, and richness indexes presented a similar trend before the administration of STT (Figure 5A, Table 4; Appendix A). All pre-treatment patients had low species diversity and richness indexes. In contrast, post-treatment patients presented with higher richness and diversity values, attributed to dysbiosis in the microbial structure and composition of the pre-treatment bacterial communities, which led to bacterial recolonization that resisted the antimicrobial treatment. Simpson’s dominance index revealed statistically non-significant values in both groups. 

The parallel PCoA plot revealed the highest variation (29.65%) among the study samples (Figure 5B). Moreover, when determining clusters in all variation axes, no correlation between pretreatment and posttreatment patients was observed, suggesting an almost unique community structure and composition in each sample. This fact was confirmed using the ANOSIM test (Table 5), where no statistical significance was observed regarding the similarities between the two study groups. These results confirmed the multiple dysbiotic events that occurred due to STT.

The paired Student’s *t* test and the Wilcoxon signed rank test were also performed to determine statistically significant differences in the bacterial diversity and richness in both pretreatment and post-treatment groups (Table 6). However, statistically non-significant results were determined between the study groups.

A heatmap of the behaviour of specific bacteria of clinical interest before and after the administration of STT (Figure 6) revealed full eradication of *Brevundimonas*, *Tetragenococcus*, and *Shewanella*. Moreover, *Pantoea*, *Klebsiella*, *Pseudomonas*, *Prevotella*, and *Acinetobacter* colonized during the resilience period. Although STT induced severe alterations, *Corynebacterium*, *Staphylococcus*, and *Cutibacterium* only diminished their relative abundances. The heatmap also revealed the clustering of the patients before and after the administration of STT, indicating the similarity of the samples, regardless of the treatment status. Four clusters were determined according to the similarity in the microbial community structure, as observed in patients 1A, 2A, and 4A; 1B, 2B, and 5B; patients 3A and 6B; and patients 4B, 5A, and 6A. However, patient 3B could not be clustered into any of these groups.

### 3.2. Correlation of C. acnes in Gastric Diseases

Forty FFPE gastric biopsy samples (Appendix A) from patients with non-significant histologic alterations, gastritis, gastropathies, and intestinal metaplasia were evaluated using qRT-PCR. Figure 7 shows the correlation between the presence of *C. acnes* and *H. pylori* in the FFPE samples obtained from patients with gastric infectious diseases. Seventeen patients revealed *C. acnes* transcripts, whereas *H. pylori* was determined as absent. Five patients with non-significant alterations were exclusively colonized with *C. acnes*. However, when evaluating gastropathies, six samples were presented with both bacteria. The number of *C. acnes* transcripts was significantly higher than that of *H. pylori* transcripts. Finally, when evaluating the samples from patients diagnosed with intestinal metaplasia, only one patient presented with *C. acnes* transcripts. The remaining patients did not present *C. acnes* or *H. pylori* transcripts (Appendix A).

As observed in Table 7, the number of *C. acnes* transcripts was determined as statistically significant compared with the number of *H. pylori* transcripts, with the non-parametric Mann–Whitney U-Test.

The number of *C. acnes* transcripts was determined significantly higher than the number of *H. pylori* transcripts (Table 8).

Additionally, the significance of the number of *C. acnes* transcripts by pathology was determined using the Kruskal–Wallis test (Table 9). As observed, the number of *C. acnes* transcripts was higher in gastritis (maximum = 573,263,008.44), whereas it was lower in gastropathies (maximum = 48,151,777.41).

## 4. Discussion

The health status of the host can be significantly altered by the induction of dysbiosis in the human gastrointestinal microbiome, which increases the susceptibility to gastric disorders [16], and difficulting both clinical diagnosis and treatment of infections.

The diagnosis for the identification of *H. pylori* infection is not easy [36]. Therefore, different methodologies are necessary for clinical diagnosis. Nevertheless, each technique presents variable sensitivity and specificity. Histopathological evaluation revealed that patients 3B and 4B tested positive for *H. pylori*. However, microbiological and molecular analyses only confirmed patients 1B, 5B, and 6B as *H. pylori* positive. Metagenomics revealed *H. pylori* representative sequences in patients 3B (interpreted as an occult infection event), 5B, and 6B, but not in patient 1B. Although this may be interpreted as a false-negative result, we attribute this result to the sampling site. Conflicting reports regarding the best sampling site for the detection of *H. pylori* continue because of the migration mechanisms of this bacterium in the stomach [37,38]. Advanced stages of histological injury have been also implied in the diminishment of *H. pylori* populations, reducing its detection rates [39] and hindering its identification using molecular methods, including metagenomics.

Coccoid structures were detected in patient 5A via histopathological evaluation. Although *H. pylori* can modify its native morphology under dysbiotic microenvironmental conditions (i.e., antibiotic and antisecretory therapies, and accumulation of toxigenic metabolic products (reactive O_2_ species, pyrimidine nucleotides)) as an adaptative mechanism [40,41], it was impossible to conclude its presence. Therefore, we attribute this finding to the muco-microbiotic layer, which is described from a morphofunctional and histological point of view, as the first line of defense under hostile conditions. This structure is the product of the union of the mucus layer and microorganisms. Although in most routine histologic evaluations the muco-microbiotic layer is unfortunately not visible due to the processing of the sample, the detection of microbial morphologies in *H. pylori*-negative subjects, as observed in our study, can elucidate a possible key role of these bacteria in the gastrointestinal physiology and pathophysiology [42]. 

Patients 1B, 6B, and 4B presented with urealytic activity. Although the urea breath test is one of the fastest and most employed tests for clinical diagnosis [43], studies have reported false-positive rates of up to 16.9% [44,45], supporting other reports that have also observed native commensal bacteria with urealytic activity in the stomach, e.g., *Staphylococcus epidermidis*, *Streptococcus salivarius*, and *Staphylococcus capitis urealiticum* [46].

Although these results strongly suggest a lack of trustworthy in the different tests for clinical diagnosis, this study attempts to highlight the importance of the integration of all diagnostic tools for the identification of *H. pylori*. We suggest not treating all results as mutually excluding but as complementary data. 

Although the international guidelines for the treatment of *H. pylori* infection usually suggest the use of the STT composed of CLR, AMX, and omeprazole [8,9], this therapy has lost its effectiveness due to the increasing resistance rates to antibiotics [47], lengthened periods of treatment up to 2 weeks, and the intention to reduce treatment periods to less than 1 week [48]. To date, STT with CLR is considered one of the least effective treatments since its eradication rates stand < 73% [49], forcing the clinical practice to reconsider other broad-spectrum antibiotics, such as MTZ, which has shown an improvement in the symptomatology and eradication rates > 94.3% when combined with AMX and omeprazole [48].

The lack of objectivity in selecting an optimal therapy according to the phenotypic and genotypic characteristics of *H. pylori* remains a major challenge, particularly with regard to the increasing rates of antimicrobial resistance, which in consequence hamper the treatment of the bacterial infection, and rapidly increases the gastroduodenal morbidity rates [50]. Three primary cultures isolated from the pretreatment patients presented with specific susceptibility profiles to the four antibiotics. However, when testing the subcultures, discrepancies were observed in their susceptibility profiles. Although the emergence of multidrug-resistant strains has been recognized as a growing problem for the treatment of these infections, heteroresistance, a non-widely discussed issue, must be examined [50].

Heteroresistance, which is defined as the different susceptibility profiles to specific antibiotics in *H. pylori* subpopulations [51], has been poorly detected mainly due to the lack of standardized methods for its characterization. However, our findings justify the need for new methods for the optimal detection and genotyping of these subpopulations prior to a multiple-antibiotic therapy prescription [50].

Clinical outcomes are attributed to strain-specific virulence factors [52]. Although *cagA*+ increases the risk of mucosal inflammation [53,54], highly diverse *vacA s1m1/s2m2* variants were correlated with more severe histological lesions and clinical outcomes [55,56]. Demirturk et al. [57] reported severe atrophy in diverse *cagA* and *vacA* genotypes, revealing a higher risk of progression of precancerous lesions than each virulence factor considered separately. Urtiz-Estrada et al. [58] identified that the *cagA+vacAs1+m1* genotype is the most prevalent genotype in Mexican patients and is correlated with various gastric diseases. In our study, it was the most observed genotype; however, some studies widely suggest the possible association of non-*H. pylori* microbiota with the induction of histological injuries in *H. pylori*-negative patients [25].

Patient 2A revealed *H. pylori* representative sequences after the administration of STT. Buffie et al. [59] reported that dysbiotic events can facilitate the establishment of infections as recent acquisition events during the recolonization period, which was determined after the administration of STT. Although some studies reported de novo infections via the inadequate sterilization of surgical materials [60,61], other factors must be considered for these types of events, such as the habits of the host during the infection treatment process, including age, sex, diet, fomites, lipid metabolism, smoking, alcohol consumption, and physical activity [62].

Although dysbiosis is mainly attributed to STT administration, the broad-spectrum activity of the antibiotics prevented the development of infections by not allowing exogenous bacteria to infiltrate the mucosal tissue during therapy, in addition to the competition mechanisms for the inhibition of colonization resistance, pressure selection, and regulation of overgrowth by the survivor gastric microbiota (production of bacteriocins, alterations in gastric pH, consumption of limited resources for competition, and promotion of the epithelial barrier by antimicrobial peptides) [63,64].

Bacterial communities that were altered during dysbiosis (Figure 7) have been reported to be involved in the maintenance of gastrointestinal homeostasis, e.g., signalling for the release of gastric acids [65]; regulation of pathobiont overgrowth [66]; synthesis of precursors in the synthesis of short-chain fatty acids [67]; metabolism of processed foods and production of histamine under halophilic conditions [68]; immunomodulation [69,70,71,72]; and lipid digestion processes [73]. Some of these bacteria have also been shown to be associated with secondary infections in immunocompromised patients [74,75,76,77,78,79,80]. 

Diversity and richness index values after the recolonization period varied between individuals. The overgrowth and dominance of pathobiont bacteria of clinical interest were observed; however, both metrics after the administration of STT slightly increased, suggesting recolonization which we attribute to the lifestyle [16]. Recolonization by pathobionts was observed. Palleja et al. [81] observed recolonization of the intestinal microbiota after the administration of a multiple-antibiotic therapy. Most species recovered their almost native relative abundances 42 days after treatment, suggesting the modulation of recovery patterns by antibiotics resistance genes (ARGs). In our study, recolonization was observed to be predominantly performed by non-dominant facultative bacteria on day 30. Birg, Ritz, and Lin. [64] reported that eradication treatments generate organ-specific dysbiosis by inducing oxygenation of the gastric tissue via severe inflammation processes, favouring the growth of facultative anaerobic pathogens with antibiotic resistance genes (ARGs).

Recolonization by pathobiont bacteria is an actual concern due to their ability to infect patients with an increased risk of infection. Our study revealed the dominance of *Pseudomonas* after dysbiosis over time [82]. Although this genus colonizes healthy subjects, it can overgrow on almost any surface due to its non-restrictive metabolic requirements. *Pseudomonas aeruginosa* and *Pseudomonas fluorescens* are considered to be of clinical interest because of their role as opportunist pathogens in healthcare-associated infections (HAIs) [83,84], in addition to the risk of establishment of carbapenem-resistant *P. aeruginosa* [82]. 

Microbial communities with specific functions (the inhibition of *H. pylori* growth and its conversion to coccoid structures via the modulation of uremic toxins [18]; acquisition and competition of nutrients to prevent the establishment of *Escherichia coli* pathotypes [85,86,87,88]; bioeradication and recovery from infectious diseases [89,90]; and generation of energy [91,92]) increased their relative abundances in the gastric microenvironment of the study subjects. However, these bacteria have been also associated with immunocompromised patients and those with gastric diseases [85,86,87,88,91,92,93,94,95].

According to the results of our study, we strongly suggest that dysbiotic events, such as bacterial eradication by the administration of a multiple-antibiotic treatment, and its consequences, e.g., survival and overgrowth of adapted microbial communities, indicate a true ecological opportunity for these pathobionts to persist in the microenvironment by benefitting of the alterations induced in the native microbiota, e.g., eradication of non-resistant native microbiota, whose principal functions might include the regulation of the establishment or overgrowth of specific pathobionts or exogenous microbiota, both of clinical interest. 

All pre-treatment patients were initially diagnosed with *H. pylori* infection. Histological injury was observed as well. However, only three patients showed relative abundances < 9% and <1% of this bacterium. Both presence and bacterial overgrowth are widely correlated with the pathogenesis of infectious diseases, which were not observed for *H. pylori* in this study. Through metagenomics, *C. acnes* was observed to be dominant in almost all pre-treatment patients, regardless of the presence–absence of histological injury (i.e., patient 2), strongly suggesting the role of non-*H. pylori* microbiota in not only development but also the initiation of gastric infectious diseases [96].

To support our findings, FFPE gastric biopsy samples were evaluated. *H. pylori* was exclusively present in samples from patients with gastropathies. This bacterium is the most frequent cause of gastroduodenal diseases due to its evident dominant relative abundances in subjects with these conditions [97]. However, *H. pylori* has coevolved within humans through time as a pathobiont by inducing multiple benefits within the host [98]. Additionally, studies have reported the incidence of gastric alterations in the absence of this bacterium in worldwide populations, in which the aetiology of the cases could not be determined [99,100], suggesting a possible role of specific gastric bacteria under dysbiosis. 

Research on the microbiome and its significant findings regarding the dysbiosis of several clinical outcomes opens the opportunity to study other bacterial agents possibly involved in the initiation and development of infectious diseases [101]. Studies have reported an overgrowth of *Paludibacter* sp. and *Dialister* sp. in patients with gastritis [102], whereas other authors have observed the role of an overabundance of *Streptococcus* spp., *Haemophilus parainfluenzae*, and *Treponema* spp. in the development and progression of dysbiosis in patients with non-*H. pylori* gastritis [103], suggesting new potential associations between the absence of this pathobiont and the development of infectious diseases [101]. Native commensal bacteria (i.e., *Lactobacillus* spp., *Prevotella melaninogenica*, *Streptococcus anginosus*) have also been shown to be associated with the development of peptic ulcer and gastric cancer [17,18]. The skin pathobiont *Cutibacterium acnes* was present in almost all FFPE samples. 

Neither *H. pylori* nor *C. acnes* transcripts were identified in intestinal metaplasia gastric biopsy samples, except in one patient. Studies have determined significant differences in gastric microbial diversity, which is gradually reduced while progressing from non-atrophic gastritis to gastric cancer [104] due to an increase in the production of proinflammatory cytokines and subsequent progressive inflammation [105]. Although these conditions result in an inhospitable microenvironment for most native bacteria, studies have reported dysregulation of the bacterial overgrowth of lactic acid bacteria, which can also promote the development of neoplasia [101,105]. Patient 30 was colonized with *C. acnes*, which has shown overabundance in gastric tumoral tissues because of its ability to induce inflammation via the production of interleukin-15 [105]. 

## 5. Conclusions

NGS tools enable us to determine alterations in the microbiota under specific health-disease conditions. The dysbiotic events of specific pathobionts were highlighted by the dominance of *C. acnes* in the gastric microenvironment, suggesting the possible role of this bacterium in the initiation or development of diseases. However, some limitations could not allow us to conclude its possible role in the stomach. To support our findings, the presence of *C. acnes* was evaluated, allowing us to highlight its dominance in different gastric alterations. Therefore, gastroduodenal disorders should no longer be considered as self-limiting diseases. Studies regarding the characterization of *C. acnes* must be conducted to evaluate the functions of this bacterium in the gastric microenvironment and determine its role in the pathogenesis of infectious diseases. This study takes part in the list of studies that have characterized dysbiotic events, in addition to the recent emergence of pathobiont bacterial species, which under a disequilibrium state can induce severe injury to the host, instead of generating multiple benefits to a specific organ.

## Figures and Tables

**Figure 1 microorganisms-11-02748-f001:**
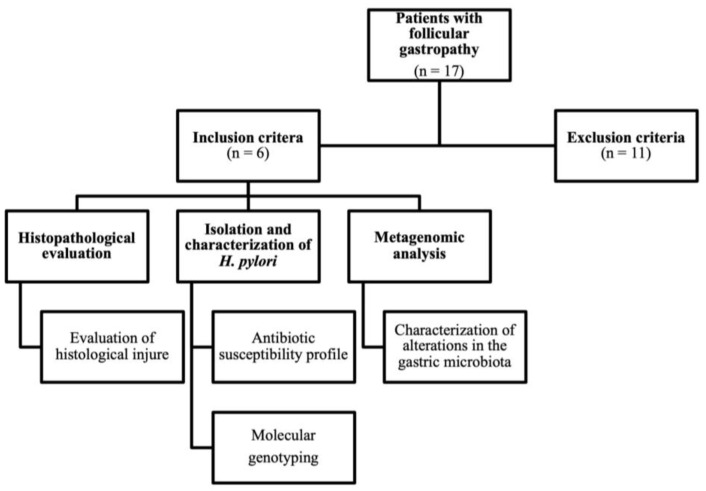
Patient enrolment flowchart.

**Figure 2 microorganisms-11-02748-f002:**
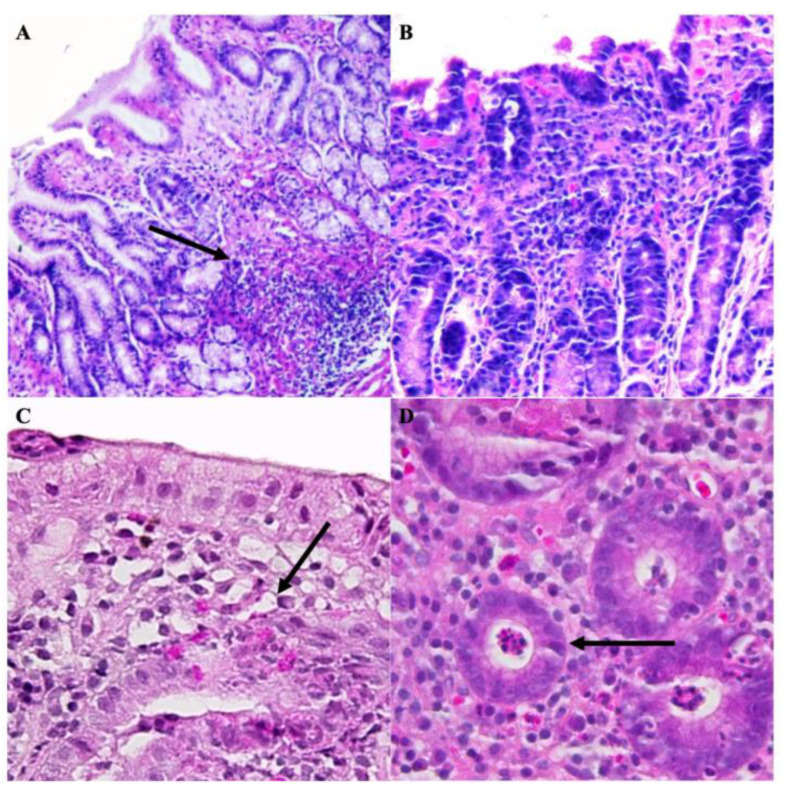
Histological findings in the gastric mucosal tissue of patients with follicular gastropathy and pangastropathy. (**A**) Lymphoid follicles are highlighted with ↑. Diminishment of glands (H&E; 10×). (**B**) Mononuclear cells and glandular regenerative changes (H&E; 10×). (**C**) Active destruction of glands by cellular infiltrates (neutrophils, eosinophils, and mononuclear cells; highlighted with ↑) (H&E; 40×). (**D**) Glandular destruction, diffuse combined inflammatory infiltrates, and intraluminal microabscessation (H&E; 40×).

**Figure 3 microorganisms-11-02748-f003:**
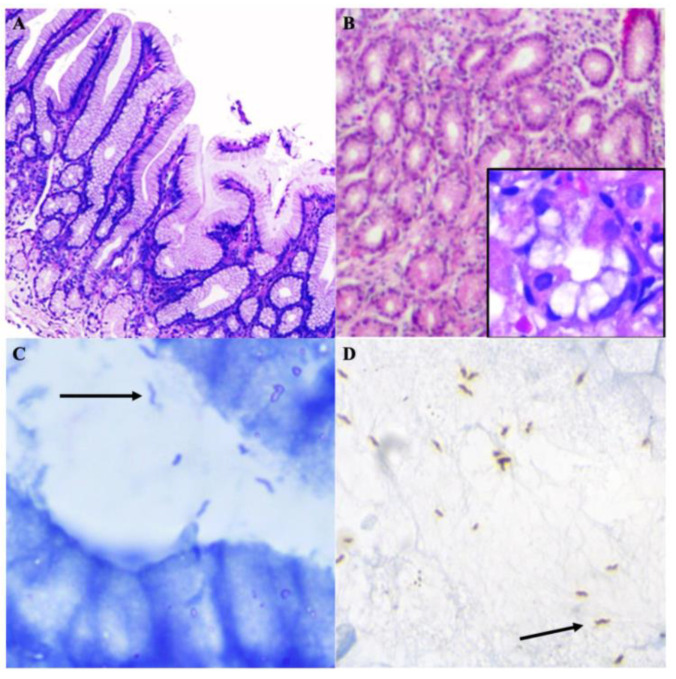
Histological injury and detection *of H. pylori* in the gastric mucosal tissue of patients with follicular gastropathy and pangastropathy. (**A**) Prominent foveolar hyperplasia determined in > 50% of total length mucosa (H&E; 10×). (**B**) Atrophy, inflammatory alterations, and goblet cells (H&E; 40×). (**C**) Typical bacillary structures of *H. pylori* (highlighted with ↑) observed using the Giemsa staining technique (H&E; 10×). (**D**) Detection of *H. pylori* using immunohistochemical staining (H&E; 10×).

**Figure 4 microorganisms-11-02748-f004:**
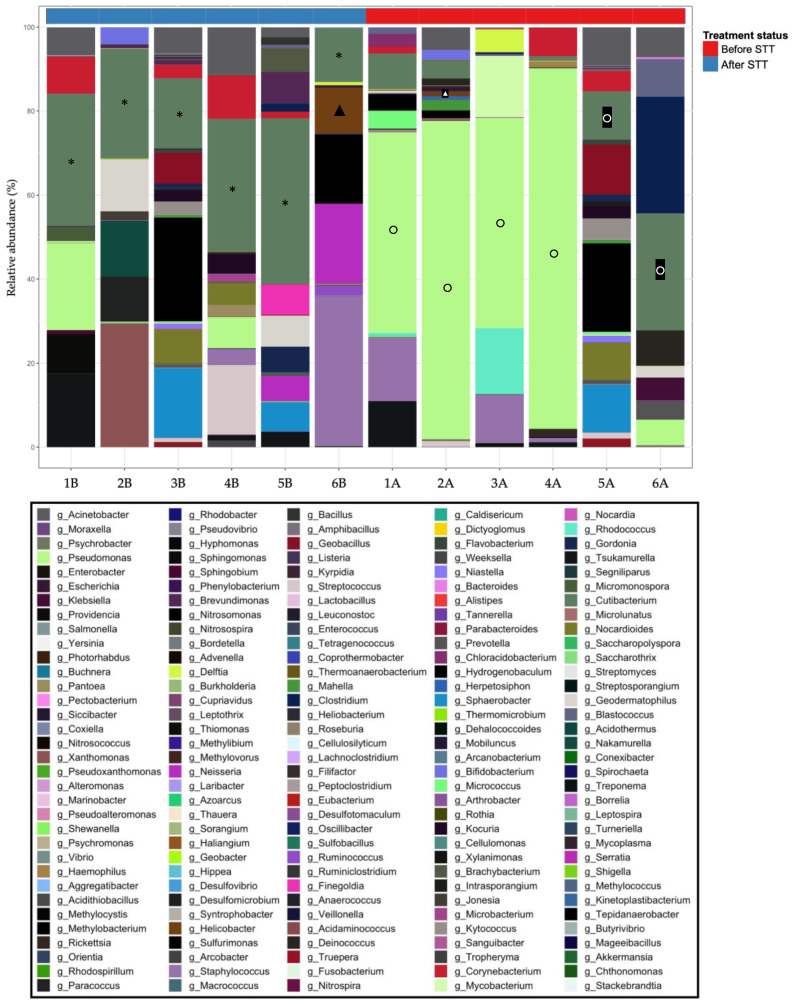
Metagenomic profile at the genus level of patients diagnosed with follicular gastropathy and pangastropathy, initially associated with *H. pylori* infection. The presence of *C. acnes* is observed in all pre-treatment patients (highlighted with *). *H. pylori* was identified in relative abundances <9% (highlighted with ▲) and <1% (not shown). After the administration of STT, overgrowth of the pathobiont bacteria of clinical interest was observed, including *Pseudomonas* (highlighted with ○). A reduction in the relative abundances of *Cutibacterium* due to STT was identified as well (highlighted with 
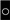
). *Helicobacter* representative sequences were observed in a posttreatment patient (highlighted with 
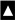
), B: Pre-treatment patient. A: Post-treatment patient.

**Figure 5 microorganisms-11-02748-f005:**
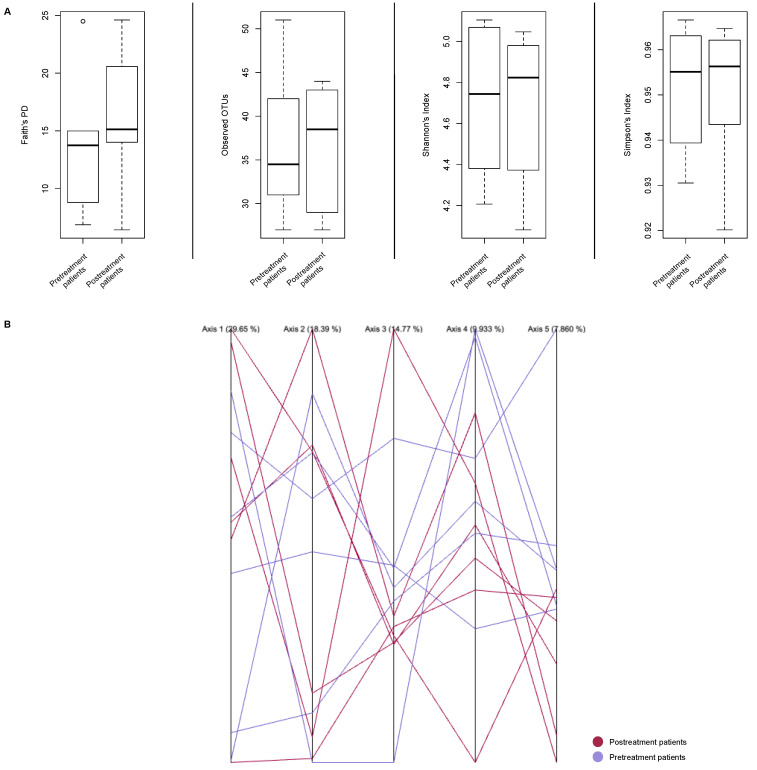
Alpha- and Beta-diversity indexes. (**A**) Alpha diversity estimations in pre-treatment and post-treatment patients. (**B**) Parallel Principal Coordinate Analysis (PCoA) indicates the variation between the pre-treatment and post-treatment microbiota. Axis 1 variation value = 29.65%; Axis 2 variation value = 18.39%; Axis 3 variation value = 14.77%; Axis 4 variation value = 9.933%; Axis 5 variation value = 7.860%.

**Figure 6 microorganisms-11-02748-f006:**
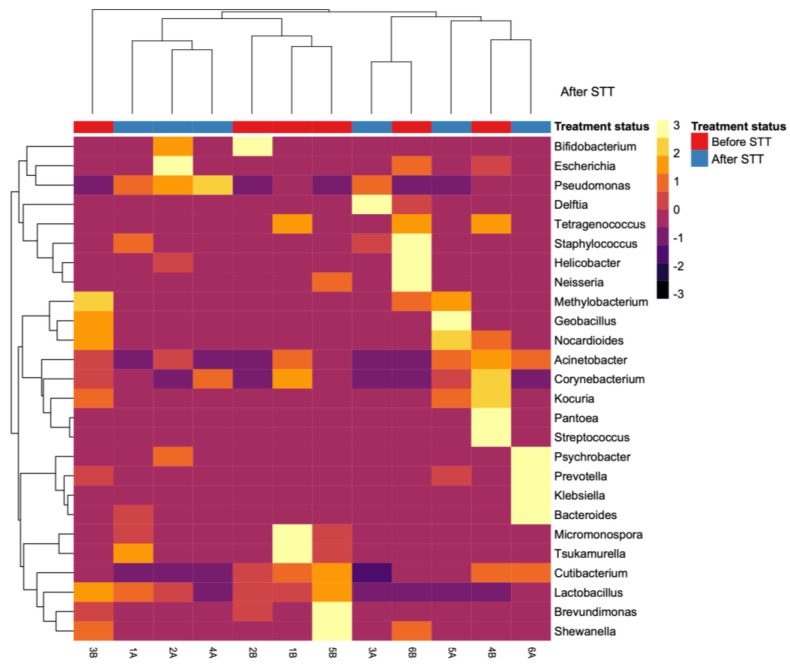
Effect of the STT on the gastric microbial communities of clinical interest. The heatmap highlights specific bacteria of clinical interest (rows) and the alterations of their relative abundances after the administration of the STT. Metagenomic profiles were clustered according to their own microbial community structure. B: Pre-treatment patient. A: Post-treatment patient.

**Figure 7 microorganisms-11-02748-f007:**
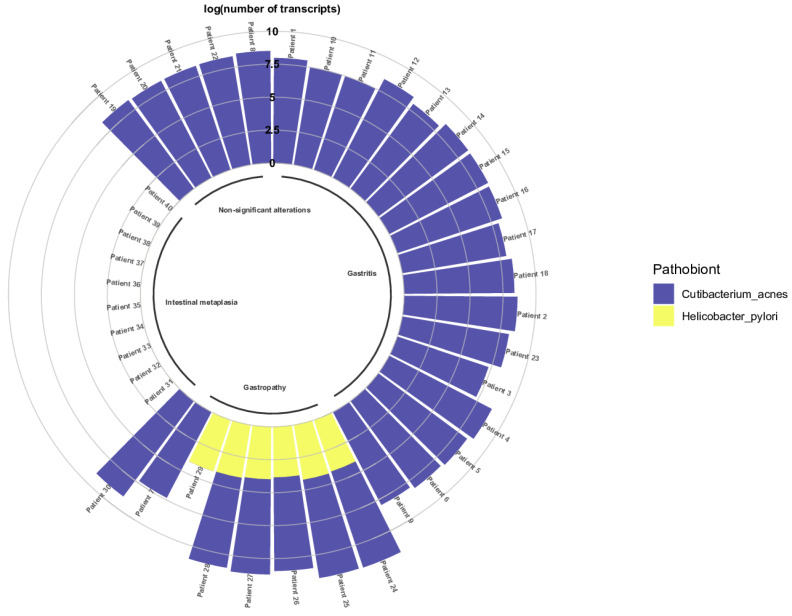
Presence of *C. acnes* and *H. pylori* in patients with gastric diseases. Evaluation of the presence of *C. acnes* and *H. pylori* in FFPE gastric biopsies via the determination of the log of the number of transcripts of both bacteria in the gastric biopsy samples.

**Table 1 microorganisms-11-02748-t001:** Histopathological evaluation.

	Patients											
Category	1B	1A	2B	2A	3B	3A	4B	4A	5B	5A	6B	6A
Atrophy	1	0	0	0	0	0	2	0	0	1	1	0
Mononuclear infiltrates	2	1	0	0	1	1	3	2	2	3	1	1
Neutrophilic infiltrates	0	0	0	0	0	0	3	1	0	1	0	1
Metaplasia	0	0	0	0	0	0	0	0	0	1	0	0
Lymphoid follicles	0	0	0	0	2	0	3	1	0	1	0	0
Glandular atrophy	0	0	0	0	0	1	1	1	0	0	1	0
Foveolar hyperplasia	2 60%	0	0	0	0	0	2 50%	2 60%	0	1 30%	1 30%	1 30%
Regenerative changes	1	1	0	0	0	1	2	1	0	1	0	0
Bacillary structures	0	0	0	0	1	0	1	0	0	1 *	0	0

B: Pre-treatment patient; A: Post-treatment patient. 0: Absent injure; 1: Minimum injury; 2: Moderate injury; 3: Severe injury. *: Coccoid forms.

**Table 2 microorganisms-11-02748-t002:** Identification of *H. pylori* by microbiological and molecular methods.

Patient	PCR				Urease	
	*glmM*		*cagA*	*vacA*		
	Before STT	After STT			Before STT	After STT
1	+ *^,c^	−	+ ^c^	+ ^c^	+ *^,c^	−
2	−	−	−	−	−	−
3	−	−	−	−	−	−
4	−	−	−	−	+ *^,c^	−
5	+ *^,c^	−	+ ^c^	+ ^c^	+ *^,c^	−
6	+ *^,c^	−	+ ^c^	+ ^c^	−	−

+: Positive test. −: Negative test. ^C^: Identification from culture. *: Identification from gastric biopsy.

**Table 3 microorganisms-11-02748-t003:** Estimation of Good’s coverage and Observed OTUs indexes.

Patient	Good’s Coverage		Observed OTUs	
	B	A	B	A
1	1	1	27	27
2	1	1	34	40
3	1	1	42	29
4	1	1	35	37
5	1	1	51	44
6	1	1	31	43

B: Pre-treatment patient. A: Post-treatment patient.

**Table 4 microorganisms-11-02748-t004:** Alpha indexes.

Patient	Shannon’s (H)		Faith’s PD		Chao1		Simpson’s Index	
	B	A	B	A	B	A	B	A
1	4.207	4.373	6.861	14.009	27	27	0.931	0.944
2	4.633	4.979	8.802	20.566	34	40	0.951	0.965
3	5.068	4.082	24.488	6.423	42	29	0.967	0.920
4	4.853	4.679	14.987	14.856	35	37	0.959	0.953
5	5.104	5.046	14.097	24.589	51	44	0.963	0.959
6	4.381	4.968	13.394	15.409	31	43	0.939	0.962

B: Pre-treatment patient. A: Post-treatment patient.

**Table 5 microorganisms-11-02748-t005:** Analysis of similarity (ANOSIM) test.

Group 1	Group 2	n	Permutations	R	*p*	*q*
B	A	12	999	0.073	0.273	0.273

B: Pre-treatment patients. A: Post-treatment patients.

**Table 6 microorganisms-11-02748-t006:** Paired Student’s *t* test and Wilcoxon signed rank test for beta-diversity analyses.

	Good’s Coverage	Observed OTUs	Shannon’s (H)	Faith’s PD	Chao1	Simpson’s Index
Paired *t* test *p*-value	1	1	0.914	0.643	1	0.933
Wilcoxon (W) *p*-value	1	0.81	0.749	0.423	0.81	0.936

**Table 7 microorganisms-11-02748-t007:** Significance of the number of *C. acnes* and *H. pylori* transcripts.

Metric	Pathobiont	
	*C. acnes*	*H. pylori*
Mean	192,315,183.00	11,538.67
Standard deviation	203,649,007.30	3168.79
n	29	6
Median	157,942,606.40	11,973.67
P25	36,511,873.51	9478.76
P75	230,486,322.70	12,922.57

n: sample size; *p*-value < 0.001.

**Table 8 microorganisms-11-02748-t008:** Determination of the significance between the number of *C. acnes* and *H. pylori* transcripts in the gastropathy group using the Mann–Whitney U-Test.

Number of Transcripts (n = 7)	Median	25th Percentile	75th Percentile	Minimum	Maximum	*p*-Value
*C. acnes*	16,915,313.11	8,383,738.74	36,511,872.51	0.90	48,151,777.41	0.017
*H. pylori*	11,383.78	6807.32	12,922.57	0.90	16,076.05	

n: sample size.

**Table 9 microorganisms-11-02748-t009:** Determination of the significance of the presence of *C. acnes* in the gastric pathologies.

Pathology	Median	25th Percentile	75th Percentile	Minimum	Maximum	*p*-Value
Gastritis	157,942,606.40	94,564,890.45	272,850,376.48	19,229,694.53	573,263,008.44	<0.001
Gastropathy	19,388,023.30	12,826,312.73	36,511,872.51	8,383,738.74	48,151,777.41	
Non-significant alterations	221,339,080.30	202,746,241.91	230,486,321.70	194,699,912.73	318,670,312.02	
Intestinal metaplasia	<1	<1	<1	<1	990,328,266.98	

*p*-value < 0.001.

## Data Availability

The original datasets generated and analysed for this study can be found in the EMBL’s EBI European Nucleotide Archive repository, under the accession number PRJEB60885. Submitted on 8 April 2023. LINK: [https://www.ebi.ac.uk/ena/browser/view/PRJEB60885].

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
