# Peer review of "Dysbiosis by Eradication of Helicobacter pylori Infection Associated with Follicular Gastropathy and Pangastropathy"

_microorganisms, 2023, doi:10.3390/microorganisms11112748_

Round 1

Reviewer 1 Report

Comments and Suggestions for Authors

I still do not have a clear idea of what the manuscript is about. I think the authors have done a lot of analysis but the presentation and the way the story is told do not make justice to the amount of work invested in this. Here are my specific comments

INTRODUCTION:

Lines 53-56: “Recent studies have highlighted the possible association of non-H. pylori bacteria in the initiation and development of infectious diseases [5] by correlating bacterial overgrowth in non- favorable microenvironmental conditions, even in in absence of H. pylori [6].” I don’t really understand this sentence. You mean, bacteria other than H. pylori have been associated with follicular gastropathy and pangastropathy? Please clarify in the text.

Line 57: “Nevertheless, H. pylori is still classified as a high priority bacterium [7].” Vague sentence. High priority in which sense? High priority pathogen? High priority regarding AMR? Please clarify in the text. Also, the reference used is very old (1994). Please update.

Line 65-67: “Antibiotics have revolutionized the treatment of bacterial infectious diseases, con- 65

sidered as one of the mayor contributions due to the increasement of life expectance by 66

exponentially diminishing its morbidity and mortality rates.” This sentence is grammatically wrong and needs to be rephrased.

Line 65-67: Unfortunately, the sometimes-unjustified employment and abuse of drugs for relief of gastric symptomatology or self-treatment of several infectious diseases has resulted in a global crisis of resistance to multiple antibiotics. I think the global AMR crisis is way bigger than abuse of drugs for relief of gastric symptomatology or self-treatment. Yes, it is caused by antibiotic overuse but not only in human med, but also vet med and food production so please remove or rephrase and add corresponding citations.

Line 70-71: “Although antibiotics are widely employed for eradication of specific pathogenic bacteria [12]”. Are they? Can you give me examples of disease eradication based on the application of antibiotics? To date, the World Health Organization has declared only 2 diseases officially eradicated: smallpox caused by variola virus (VARV) and rinderpest caused by the rinderpest virus (RPV). I don’t think we can thank antibiotics for the eradication of this disease. Please rephrase.

Line 72-75: “Dysbiosis, defined as the induction of several alterations in the native microbiota of the host, can be observed as a significant reduction, or permanent loss of commensal species in the affected microbial communities, and/OR the establishment, colonization or overgrowth of gastrointestinal pathobionts (native commensal microbiota that present pathogen behaviours under dysbiotic conditions) [13]. Colonization of pathobionts can happen with or without the loss of commensal species. Please rephrase and add a better citation more general and not focus to H. pylori and dysbiosis. In this sentence, authors are describing dysbiosis and have not mentioned H. pylory at all.

Line 77: “H. pylori has been intimately associated to several infectious diseases”. This sentence has no sense. There is only one infectious disease associated with H. pylori that is H. pylori infection. H. pylori infections can present different pathologies or symptomatology. Is that what the authors mean? Or, are the authors talking about coinfections with other bacteria/viruses? Please rephrase.

Line 79: “Several non-invasive diagnostic techniques have been widely developed for detection of H. pylori.” Be more specific and describe the diagnostic tools available for the detection of H. pylori in cases of follicular gastropathy and pangastropathy.

Line 80-82: “However, endoscopic studies, considered as invasive methods for clinical diagnosis, are still helpful due to it facilitates an early decision taking with regard the identification of the aetiological agent, considering that other diagnostic tests require time and resources for its preparation and interpretation [14].” Be more specific and describe the diagnostic tools available for the detection of H. pylori in cases of follicular gastropathy and pangastropathy. Also, this sentence is grammatically incorrect.  So, if it is an essential sentence please rephrase.

The tittle of the manuscript is: “Dysbiosis by eradication of Helicobacter pylori infection associated to follicular gastropathy and pangastropathy.” However, the authors do not show a clear connection between the presence of H. Pylori and this gastropathies. I would recommend the authors to include a paragraph stating how many cases of follicular gastropathy and pangastropathy are caused by H. pylori a year and how many resolve with the application of antimicrobial therapy.

Line 84: “The aim of this study is to characterize the gastric microbiota of patients with follicular gastropathy and pangastropathy, evaluate the effect of STT in the gastric environment, and correlate the presence of pathobiont C. acnes in gastric infectious diseases.” Similar to the comment above. There is no explanation about C. acnes or the type of diseases it is associated with. Please include a paragraph about it in the introduction so the readers can understand your aim.

MATERIALS AND METHODS

Line 90: “Seventeen patients attended to the Gastroenterology Service at the Hospital General

Dr. Fernando Quiroz Gutiérrez, ISSSTE, who signed the informed consent for gastric

sampling.”. This sentence does not make sense. Is the name of the hospital - Hospital General Dr. Fernando Quiroz Gutiérrez-? What ISSSTE means? Please correct. Please include the city and country for the Hospitals.

How long was the patient recruitment? Please add the dates to the main text of the manuscript.

Line 104: “Eighteen biopsies (pre-treatment samples) were firstly obtained from the endoscopic study. Six samples were processed for histopathological evaluation.” I assume is three biopsies per patient? Please add to main text.

Line 142: “for genotyping according to the conditions purposed by Mendoza Elizalde et al. [17].” Please elaborate more about the genotyping. Did you do sanger sequencing or NGS? How did you analyzed the data?

Line 145: “Twelve gastric biopsies were processed for isolation and purification of total RNA 145

with GeneAll® RiboEx, and synthesis of complementary DNA (cDNA) with the Jena Bi- 146

osciences SCRIPT cDNA Synthesis Kit.” If you were going to sequence bacterial DNA, why did you use an RNA extraction kit and then convert it to cDNA? Specially when there are commercial DNA extraction kits for microbiome studies available.

Line 156: “Representative sequences of dominant OTUs were extracted for multiple alignment in the Basic Local Alignment Search Tool (BLAST) [20] for species identification.” Why did you do that if you can reach species identification with dada2?

Line 159: “Sampling depth for estimation of alpha and beta diversity indexes was determined by rarefaction tests. Additionally, the Good’s coverage index was estimated to determine the sampling quality in each sample.” What program did you use for this? Please add the program and corresponding citation.

Line 162: “Beta diversity analyses were performed through PCoA analysis from a Bray-Curtis’s dissimilarity distance matrix.” What about alpha diversity? Please specify which index did you use for alpha diversity.

Line 180; “This process was repeated until gastric tissues were paraffin-free.” How did you know the samples were paraffin-free? Please be more specific.

RESULTS

Line 251: “C: Typical bacillary structures of H. pylori evidenced with Giemsa staining technique (H&E; 10X). D: Detection of H. pylori by immunohistochemical staining (H&E; 10X).” I don’t remember reading about Giemsa or immunohistochemical staining in the materials end methods. Please add.

Line 254: “Histological injure in patient 2A was not detected, suggesting the absence of an infectious process.” I don’t get this. Patient 2 seems to not present any histological injure before or after treatment according to table 1. So why mentioning on 2A? and why the correlation with infectious process? You have only detected bacteria in 3B and 4B, hence you have other patients that present histological injure and not bacteria… how do you connect the lack of histological injure with and infectious process then?

Line 257: “Patients 1B, 5B and 6B were H. pylori positive by PCR and culture (Table 2). The 257

three remaining patients were considered negative.” And you did not see anything in the tissue? What about 3B and 4B, you saw bacteria in the tissue. They were not H.pylori I assume? Is it common to find bacteria in tissues? Have you identified the bacterial structure in 3B and 4B?

Line 263: “The susceptibility profile of the primary cultures revealed resistance to CLR. Additionally, two isolates were resistant to AMX, LVX, and high resistant to MTZ (Supplementary Table 2). However, the subcultures presented multiple susceptibility profiles (Supplementary Table 3).” Why do you do susceptibility testing in both primary and secondary cultures? And how do you explain the discrepancies? Please add this discussion to the manuscript.

Line 272: “From 362,231 raw reads, 78,053 sequences were finally retained for metagenomic analysis.” Add the average of the number of raw reads per sample with the max. and the min. I also would like to see the rarefaction curves that you mentioned you did in the materials and methods. You can add them as a supp. figure.

Figure 5. are those all genera with relative abundances < 1%? What are the values on Y-axe, relative abundance? There is a lot going on you may want to help the reader and mark in the figure what you are explaining in the text. Maybe use arrows or highlight the genera of interest?

Line 296: “H. pylori was present in 2A in < 5%.” Is this based on the results from figure 5? Because describes genera, not species. What about the samples in which you found helicobacter before treatment?

Lines 297: “Dominance, diversity and richness estimators presented a similar behavior before

and after the administration of STT (Figure 6A, Table 4). All pre-treatment patients were determined with low species diversity and richness indexes. However, all posttreatment patients presented higher richness and diversity values. Simpson’s dominance index revealed non-statistically significant values in both groups. Alterations in the relative abundances of dominant bacteria were evident”. In line 297 you mention that estimators presented a similar behavior before and after treatment but then you finish the paragraph stating that Alterations in the relative abundances of dominant bacteria were evident. This is kind of contradictory so please explain. Also, Figure 5 shows dramatic changes the structure of the bacterial communities before and after treatment. How do you explain this is not reflected in your alpha and beta diversity analysis?

Line 309:The parallel PCoA plot (Figure 6B) determined a variation of 29.65% within all samples.” I don’t think this is completely true. The PCoA analysis says that one of the sources of variation explains the 29.65% but you do not know what that source of variation is. You can use the right metadata and figure it out. Why did you choose to show more than 2 axes? it makes the figure confusing. What conclusions do you drive from it?

Line 310: “Behavior of each sample revealed a unique community structure. However, none was considered as statistically significant.” Statistically significant in term of what? What did you compare to reach these results?

Line 311: “A clustering of the patients before and after the administration of STT is shown in Figure 6C, indicating the similarity of the samples regarding the microbial structure and composition.” Are they similar or not? And what does that mean? Why did you do this test? What new piece of information does this test bring to the table? This tree is the same that appears in your headblock right? it is redundant.

Line 316: “A heatmap of the behaviour of specific bacteria of clinical interest before and after 316

the administration of STT (Figure 7) revealed full eradication of Brevundimonas, Tetragen- 317

ococcus, and Shewanella. Moreover, Pantoea, Klebsiella, Pseudomonas, Prevotella, and Aci- 318

netobacter colonized during the resilience period.” You might one to add a small paragraph explaining why you select to look at these genera and not others. How do you see such drtastic changes here and figure 5 and do not see significant changes in alpha and beta diversity?

Line 327: “Student’s t and Mann-Whitney’s U tests were performed to determined statistically significant alterations before and after the administration of STT (Table 5).” I don’t think thius are the best test to determine significance. You samples are correlated (is the same patient before and after treatment) and also the variability between subjects might be to high to determine if there are significant changes promoted by treatment. Specially with such a small number. How would use mix effect models for this or at least a one-way ANOVA with repeated measurements.

Figure 8A and text. Please include how many samples you tested in total for each of the categories. So how many patients presented gastritis for example and how many tested positive for carrying the bacteria. Otherwise, this figure is kind of useless.

Figure 8B is cool but redundant.  Specially because the authors do not explain the results that thy obtained in detail.

Line 343: “An analysis of the metrics for the two pathobionts of study was performed. As observed in Table 7, the presence of C. acnes was determined as statistically significant compared to the presence of H. pylori, which was present in 15% of the FFPE samples.” What are the metrics? What does this mean? What were you testing is it the association of the presence of bacteria and certain disease? Please add explanation to main text.

Table 8. “Determination of the significance of the number of transcripts of C. acnes and H. pylori in the pathologies.” You have previously showed that H. pylori and C. acnes are not present in the same pathologies so you should not combine the transcripts.

Line 360: “The diagnosis to identify the infection by H. pylori is not easy.” According to who? Pleas add a reference. Are the contradictory results you found also reported before in the literature?

Line 376: “Therefore, we attribute this finding to the muco-microbiotic layer, described as a morphofunctional and histological point of view, that represents the first line of defense in hostile conditions [29].” Can you please describe the muco-microbiotic layer and why do you think it can be confused by coccoid structures? Can they be other bacteria present in the tissue?

Line 378: “Therefore, isolation and characterization of H. pylori cultures is highlighted for classification of the virulence, resistance, and determine the role of specific strains in the development of histological injury.” This sentence does not have much sense in this context. You are basically saying that because of the findings describe above (about the layer), the isolation and characterization of H. pylori cultures is highlighted for classification of the virulence, resistance, and determine the role of specific strains in the development of histological injury. Is this what you meant? Also, You use “Therefore” in two consecutive sentences so please rephrase and a add a citation if corresponds.

Line 389: “however, we suggest the association of histological findings in H. pylori negative pre-treatment patients, and their inflammatory processes with non-H. pylori microbiota [37], and the environmental factors (i.e., age, sex, lipid metabolism, smoking, alcohol consumption, 392

physical activity) [38]”. You can not suggest anything because you do not have analysis to support it. You have done a microbiome study with only 6 samples that might or might not test positive for H. pylori and you have not included any metadata regarding the environmental factors you mention.  

Line 395: “most employed tests for clinical diagnosis” Clinical diagnosis of what? Please add and be specific.

Line 396. If the urealytic activity test is not great what do you use it? What is most sensitive for the detection of H.pylori from all the test that you performed? How does that correlate to your results? Can you trust that the tissue that you said contained H. pylori does indeed contain it?

Line 411: “suggesting a de novo infection by surgical material.” This is pretty serious accusation specially after you mention that the diagnostic tests for H. pylori are not trustworthy and you lack any other result to support it.

Line 417: “Although dysbiosis is mainly attributed to STT, the broad-spectrum activity of the antibiotics prevented the development of infections by not allowing exogenous bacteria to establish in the mucosal tissue” This contradicts exactly what you mention above “dysbiotic events can facilitate the establishment of infections as recent acquisition events during recolonization”.

Line 424: “Recovery of diversity and richness after STT varied between individuals, which we attribute to the lifestyle [13]. Recolonization by pathobionts was observed.” Where can I see these results?

Line 419: “In our study, recolonization was performed by non-dominant bacteria at day 30, adequate time to recolonize the altered microenvironment by facultative bacteria.” Again, where are these results?

Line 449: “Eradication, survival and overgrowth of microbial communities after a dysbiotic event, such as a multiple antibiotic treatment, indicate a true ecological opportunity due to the severe alterations induced in the native communities, e.g., eradication of non- resistant native microbiota, whose principal functions include the regulation of establishment or overgrowth by specific pathobionts or exogenous microbiota, both of clinical interest.” Add citation.

Line 459: Through metagenomics, C. acnes was observed as dominant in almost all pre-treatment patients, strongly suggesting the role of non-H. pylori microbiota in the development of gastric infectious diseases [66]. Not clear with the results that you presented because C. acnes was also present in patients with not alterations. 

Line 481: The genus Cutibacterium is a dominant commensal bacterium in human skin [74]. Although most Cutibacterium species are adapted to inhabit the human skin [75,76], the specie C. acnes is mainly associated with the maintenance of skin homeostasis due to its multiple benefits in this organ. This specie was firstly reported in healthy gastric mucosa [77-79]; however, its functions in in the gastric microenvironment have not been fully elucidated yet. In recent years C. acnes has been classified as possible trigger of gastric clinical outcomes, such as the corpus-dominant lymphocytic gastritis [37] and has been considered as high-risk factor for initiation and development of gastric cancer [80]. I was expecting to see this in the introduction.

Comments on the Quality of English Language

There are many sentences that do not make sense. I have added specific comments in the attached file.

Author Response

We really appreciate all the detailed comments and suggestions from the reviewer for this manuscript. We have carefully answered each suggestion as follows:

I still do not have a clear idea of what the manuscript is about. I think the authors have done a lot of analysis but the presentation and the way the story is told do not make justice to the amount of work invested in this. Here are my specific comments

INTRODUCTION: 

Lines 53-56: “Recent studies have highlighted the possible association of non-H. pylori bacteria in the initiation and development of infectious diseases [5] by correlating bacterial overgrowth in non- favorable microenvironmental conditions, even in in absence of H. pylori [6].” I don’t really understand this sentence. You mean, bacteria other than H. pylori have been associated with follicular gastropathy and pangastropathy? Please clarify in the text. What we are trying to clarify in this sentence is the role of gastric bacteria in the initiation and development of several gastric diseases (including follicular gastropathy and pangastropathy) even in H. pylori-free subjects. Corrected.

Line 57: “Nevertheless, H. pylori is still classified as a high priority bacterium [7].” Vague sentence. High priority in which sense? High priority pathogen? High priority regarding AMR? Please clarify in the text. Also, the reference used is very old (1994). Please update. Corrected and updated.

Line 65-67: “Antibiotics have revolutionized the treatment of bacterial infectious diseases, con- 65

sidered as one of the mayor contributions due to the increasement of life expectance by 66

exponentially diminishing its morbidity and mortality rates.” This sentence is grammatically wrong and needs to be rephrased. Corrected

Line 65-67: Unfortunately, the sometimes-unjustified employment and abuse of drugs for relief of gastric symptomatology or self-treatment of several infectious diseases has resulted in a global crisis of resistance to multiple antibiotics. I think the global AMR crisis is way bigger than abuse of drugs for relief of gastric symptomatology or self-treatment. Yes, it is caused by antibiotic overuse but not only in human med, but also vet med and food production so please remove or rephrase and add corresponding citations. Rephrased

Line 70-71: “Although antibiotics are widely employed for eradication of specific pathogenic bacteria [12]”. Are they? Can you give me examples of disease eradication based on the application of antibiotics? To date, the World Health Organization has declared only 2 diseases officially eradicated: smallpox caused by variola virus (VARV) and rinderpest caused by the rinderpest virus (RPV). I don’t think we can thank antibiotics for the eradication of this disease. Please rephrase. Rephrased

Line 72-75: “Dysbiosis, defined as the induction of several alterations in the native microbiota of the host, can be observed as a significant reduction, or permanent loss of commensal species in the affected microbial communities, and/OR the establishment, colonization or overgrowth of gastrointestinal pathobionts (native commensal microbiota that present pathogen behaviours under dysbiotic conditions) [13]. Colonization of pathobionts can happen with or without the loss of commensal species. Please rephrase and add a better citation more general and not focus to H. pylori and dysbiosis. In this sentence, authors are describing dysbiosis and have not mentioned H. pylory at all. The idea has been updated.

Line 77: “H. pylori has been intimately associated to several infectious diseases”. This sentence has no sense. There is only one infectious disease associated with H. pylori that is H. pylori infection. H. pylori infections can present different pathologies or symptomatology. Is that what the authors mean? Or, are the authors talking about coinfections with other bacteria/viruses? Please rephrase. Rephrased.

Line 79: “Several non-invasive diagnostic techniques have been widely developed for detection of H. pylori.” Be more specific and describe the diagnostic tools available for the detection of H. pylori in cases of follicular gastropathy and pangastropathy. Rephrased and corrected.

Line 80-82: “However, endoscopic studies, considered as invasive methods for clinical diagnosis, are still helpful due to it facilitates an early decision taking with regard the identification of the aetiological agent, considering that other diagnostic tests require time and resources for its preparation and interpretation [14].” Be more specific and describe the diagnostic tools available for the detection of H. pylori in cases of follicular gastropathy and pangastropathy. Also, this sentence is grammatically incorrect.  So, if it is an essential sentence please rephrase. Rephrased and corrected.

The tittle of the manuscript is: “Dysbiosis by eradication of Helicobacter pylori infection associated to follicular gastropathy and pangastropathy.” However, the authors do not show a clear connection between the presence of H. Pylori and this gastropathies. I would recommend the authors to include a paragraph stating how many cases of follicular gastropathy and pangastropathy are caused by H. pylori a year and how many resolve with the application of antimicrobial therapy. Information updated.

Line 84: “The aim of this study is to characterize the gastric microbiota of patients with follicular gastropathy and pangastropathy, evaluate the effect of STT in the gastric environment, and correlate the presence of pathobiont C. acnes in gastric infectious diseases.” Similar to the comment above. There is no explanation about C. acnes or the type of diseases it is associated with. Please include a paragraph about it in the introduction so the readers can understand your aim. Paragraph from the discussion relocated to introduction.

MATERIALS AND METHODS

Line 90: “Seventeen patients attended to the Gastroenterology Service at the Hospital General 

Dr. Fernando Quiroz Gutiérrez, ISSSTE, who signed the informed consent for gastric

sampling.”. This sentence does not make sense. Is the name of the hospital - Hospital General Dr. Fernando Quiroz Gutiérrez-? What ISSSTE means? Please correct. Please include the city and country for the Hospitals. Corrected

How long was the patient recruitment? Please add the dates to the main text of the manuscript. Information added to the main text

Line 104: “Eighteen biopsies (pre-treatment samples) were firstly obtained from the endoscopic study. Six samples were processed for histopathological evaluation.” I assume is three biopsies per patient? Please add to main text. Information added to the text.

Line 142: “for genotyping according to the conditions purposed by Mendoza Elizalde et al. [17].” Please elaborate more about the genotyping. Did you do sanger sequencing or NGS? How did you analyzed the data? Information added to the main text.

Line 145: “Twelve gastric biopsies were processed for isolation and purification of total RNA 145

with GeneAll® RiboEx, and synthesis of complementary DNA (cDNA) with the Jena Bi- 146

osciences SCRIPT cDNA Synthesis Kit.” If you were going to sequence bacterial DNA, why did you use an RNA extraction kit and then convert it to cDNA? Specially when there are commercial DNA extraction kits for microbiome studies available. We sequenced cDNA instead of dsDNA to characterize the metabolically active bacteria. From these samples, the sequencing procedure was focused for exclusively targeting bacterial 16S rRNA gene. Not a metagenomic shotgun sequencing.

Line 156: “Representative sequences of dominant OTUs were extracted for multiple alignment in the Basic Local Alignment Search Tool (BLAST) [20] for species identification.” Why did you do that if you can reach species identification with dada2? The DADA2 algorithm, which is also included in the QIIME2 pipeline, was exclusively employed for quality control. Specie identification was performed with the plugin feature-classifier included in QIIME2. When optimizing the classifier for the QIIME2 pipeline, we observed at 99% of taxonomic identity, that most representative sequences were not fully annotated. For this reason, we also reduced the percentage at 97%, which allow us to identify bacterial species at Genus level.

Additionally, when we observed the relative abundances of Cutibacterium and Helicobacter spp, we decided to extract all the sequences which were previously identified as the target bacterium and aligned in the BLAST tool.

Line 159: “Sampling depth for estimation of alpha and beta diversity indexes was determined by rarefaction tests. Additionally, the Good’s coverage index was estimated to determine the sampling quality in each sample.” What program did you use for this? Please add the program and corresponding citation. Estimation of both alpha and beta diversity indexes were all determined in the QIIME2 pipeline.

Line 162: “Beta diversity analyses were performed through PCoA analysis from a Bray-Curtis’s dissimilarity distance matrix.” What about alpha diversity? Please specify which index did you use for alpha diversity. Corrected

Line 180; “This process was repeated until gastric tissues were paraffin-free.” How did you know the samples were paraffin-free? Please be more specific. We considered the tissues as paraffin-free at the second repeat of the process mentioned in the manuscript due to 1) the thick of the samples (approximately 10 um each) and 2) the fusion temperature of the paraffin (between 40-60ºC), allowing to fully deparaffinate the tissues. When observing the tissues in Xylene, paraffin was completely dissolved in Xylene. When a tissue is not properly deparaffined, miscelles can be observed at the top of the tissue in presence of ethanol or DEPC H2O.

RESULTS

Line 251: “C: Typical bacillary structures of H. pylori evidenced with Giemsa staining technique (H&E; 10X). D: Detection of H. pylori by immunohistochemical staining (H&E; 10X).” I don’t remember reading about Giemsa or immunohistochemical staining in the materials end methods. Please add. Information added to the main text.

Line 254: “Histological injure in patient 2A was not detected, suggesting the absence of an infectious process.” I don’t get this. Patient 2 seems to not present any histological injure before or after treatment according to table 1. So why mentioning on 2A? and why the correlation with infectious process? You have only detected bacteria in 3B and 4B, hence you have other patients that present histological injure and not bacteria… how do you connect the lack of histological injure with and infectious process then?  Your asseveration is correct. The idea has been removed.

Line 257: “Patients 1B, 5B and 6B were H. pylori positive by PCR and culture (Table 2). The 257

three remaining patients were considered negative.” And you did not see anything in the tissue? Information regarding histopathological evaluation has been added to 3.1.2 What about 3B and 4B, you saw bacteria in the tissue. They were not H.pylori I assume? Correct. Bacterial structures were observed in these patients. However, H. pylori was only detected in 3B. In the discussion we suggest an occult infection that was only detected through metagenomics. Is it common to find bacteria in tissues? It is not normal due to the processing of the tissues, where soluble factors and bacteria of the muco-microbiotic layer are removed. Have you identified the bacterial structure in 3B and 4B?

Line 263: “The susceptibility profile of the primary cultures revealed resistance to CLR. Additionally, two isolates were resistant to AMX, LVX, and high resistant to MTZ (Supplementary Table 2). However, the subcultures presented multiple susceptibility profiles (Supplementary Table 3).” Why do you do susceptibility testing in both primary and secondary cultures? And how do you explain the discrepancies? Please add this discussion to the manuscript. Information updated in the discussion.

Line 272: “From 362,231 raw reads, 78,053 sequences were finally retained for metagenomic analysis.” Add the average of the number of raw reads per sample with the max. and the min. I also would like to see the rarefaction curves that you mentioned you did in the materials and methods. You can add them as a supp. figure. Information added into the text. Rarefaction curves (per patient and per treatment status) added as suppl. material.

Figure 5. are those all genera with relative abundances < 1%? What are the values on Y-axe, relative abundance? There is a lot going on you may want to help the reader and mark in the figure what you are explaining in the text. Maybe use arrows or highlight the genera of interest? Figure corrected.

Line 296: “H. pylori was present in 2A in < 5%.” Is this based on the results from figure 5? Because describes genera, not species. What about the samples in which you found helicobacter before treatment? Indeed, these results are based on figure 5. H. pylori was found in patients 6B (<11%), 3B and 5B (<1% in both cases). A more detailed description has been added.

Lines 297: “Dominance, diversity and richness estimators presented a similar behavior before

and after the administration of STT (Figure 6A, Table 4). All pre-treatment patients were determined with low species diversity and richness indexes. However, all posttreatment patients presented higher richness and diversity values. Simpson’s dominance index revealed non-statistically significant values in both groups. Alterations in the relative abundances of dominant bacteria were evident”. In line 297 you mention that estimators presented a similar behavior before and after treatment but then you finish the paragraph stating that Alterations in the relative abundances of dominant bacteria were evident. This is kind of contradictory so please explain. Also, Figure 5 shows dramatic changes the structure of the bacterial communities before and after treatment. How do you explain this is not reflected in your alpha and beta diversity analysis? The paragraph has been reordered and rephrased.

Line 309: “The parallel PCoA plot (Figure 6B) determined a variation of 29.65% within all samples.” I don’t think this is completely true. The PCoA analysis says that one of the sources of variation explains the 29.65% but you do not know what that source of variation is. You can use the right metadata and figure it out. Information updated. Why did you choose to show more than 2 axes? it makes the figure confusing. What conclusions do you drive from it? Although it is known that the first two-three axes show the most important information regarding the dissimilarities in the microbial structure and composition, the reason to employ more than two axes was to show all variation percentages in different dimensions.

Line 310: “Behavior of each sample revealed a unique community structure. However, none was considered as statistically significant.” Statistically significant in term of what? What did you compare to reach these results? Information has been corrected.

Line 311: “A clustering of the patients before and after the administration of STT is shown in Figure 6C, indicating the similarity of the samples regarding the microbial structure and composition.” Are they similar or not? And what does that mean? Why did you do this test? What new piece of information does this test bring to the table? This tree is the same that appears in your headblock right? it is redundant. The tree has been removed, and the corresponding results has been added in figure 7.

Line 316: “A heatmap of the behaviour of specific bacteria of clinical interest before and after 316

the administration of STT (Figure 7) revealed full eradication of BrevundimonasTetragen- 317

ococcus, and Shewanella. Moreover, PantoeaKlebsiellaPseudomonasPrevotella, and Aci- 318

netobacter colonized during the resilience period.” You might one to add a small paragraph explaining why you select to look at these genera and not others. How do you see such drtastic changes here and figure 5 and do not see significant changes in alpha and beta diversity? As you mention above, drastic alterations were observed in the figures 5 and 7. Nevertheless, the eradicated communities by STT were somehow replaced by communities which were in low relative abundances during the recolonization period (30 days), where appearance and increase of new and remaining genera after STT was observed, respectively, slightly increasing both diversity and richness values, as observed in the rarefaction curves (supplementary material figures).

Line 327: “Student’s and Mann-Whitney’s tests were performed to determined statistically significant alterations before and after the administration of STT (Table 5).” I don’t think thius are the best test to determine significance. You samples are correlated (is the same patient before and after treatment) and also the variability between subjects might be to high to determine if there are significant changes promoted by treatment. Specially with such a small number. How would use mix effect models for this or at least a one-way ANOVA with repeated measurements. Due to the samples are correlated (by treatment status), student’s t test was employed. What we tried to highlight with this test was the differences of the b-diversity distances between the two study groups.

On the other hand, we employed the M-W test because of the same reason you mention, the small number of samples involved in the study.

Both results were consistent. This information has been relocated and updated in the main text.

Figure 8A and text. Please include how many samples you tested in total for each of the categories. So how many patients presented gastritis for example and how many tested positive for carrying the bacteria. Otherwise, this figure is kind of useless. + Figure 8B is cool but redundant.  Specially because the authors do not explain the results that thy obtained in detail. Figure 8 corrected. Figure 8A was removed and the required information was added in figure 8B. The explanation of the results was added in the corresponding text.

Line 343: “An analysis of the metrics for the two pathobionts of study was performed. As observed in Table 7, the presence of C. acnes was determined as statistically significant compared to the presence of H. pylori, which was present in 15% of the FFPE samples.” What are the metrics? What does this mean? What were you testing is it the association of the presence of bacteria and certain disease? Please add explanation to main text. Corrected.

Table 8. “Determination of the significance of the number of transcripts of C. acnes and H. pylori in the pathologies.” You have previously showed that H. pylori and C. acnes are not present in the same pathologies so you should not combine the transcripts. The name of the table was wrong. What was determined was the significance of the presence of C. acnes per pathology. It has been corrected.

Line 360: “The diagnosis to identify the infection by H. pylori is not easy.” According to who? Pleas add a reference. Are the contradictory results you found also reported before in the literature? Reference added. Absolutely, several studies have highlighted the contradictions when comparing all diagnostic tools for identification of H. pylori from different biopsies, e.g., stool, feces, gastric biopsies. In the manuscript, we discuss the migration events of this bacterium to other regions of the stomach, mainly due to the alterations of the microenvironmental gastric conditions, which can be induced by PPIs, abuse of antibiotics), leading to a possible false negative result in the diagnosis. However, these topics are not widely developed and reported.

Line 376: “Therefore, we attribute this finding to the muco-microbiotic layer, described as a morphofunctional and histological point of view, that represents the first line of defense in hostile conditions [29].” Can you please describe the muco-microbiotic layer and why do you think it can be confused by coccoid structures? Can they be other bacteria present in the tissue? Information updated.

Line 378: “Therefore, isolation and characterization of H. pylori cultures is highlighted for classification of the virulence, resistance, and determine the role of specific strains in the development of histological injury.” This sentence does not have much sense in this context. You are basically saying that because of the findings describe above (about the layer), the isolation and characterization of H. pylori cultures is highlighted for classification of the virulence, resistance, and determine the role of specific strains in the development of histological injury. Is this what you meant? Also, You use “Therefore” in two consecutive sentences so please rephrase and a add a citation if corresponds. Corrected.

Line 389: “however, we suggest the association of histological findings in H. pylori negative pre-treatment patients, and their inflammatory processes with non-H. pylori microbiota [37], and the environmental factors (i.e., age, sex, lipid metabolism, smoking, alcohol consumption, 392

physical activity) [38]”. You can not suggest anything because you do not have analysis to support it. You have done a microbiome study with only 6 samples that might or might not test positive for H. pylori and you have not included any metadata regarding the environmental factors you mention. The idea has been corrected.

Line 395: “most employed tests for clinical diagnosis” Clinical diagnosis of what? Please add and be specific. Corrected

Line 396. If the urealytic activity test is not great what do you use it? What is most sensitive for the detection of H.pylori from all the test that you performed? How does that correlate to your results? Can you trust that the tissue that you said contained H. pylori does indeed contain it? This is exactly what we are trying to remark when discussing our results. At the present time, there is no standardized technique with 100% sensitivity and specificity. Therefore, we added a paragraph where we try to focus our results and discussion to a possible solution (the integration of all the results obtained from all the tests employed in the routine evaluation for clinical diagnosis).

Regarding the reliability of the gastric samples evaluated in this study and talking in general. Although the standardized protocols for gastric sampling indicate a 100% sensitivity for ID of H. pylory, we cannot be 100% sure the obtained biopsy was the correct to get a final diagnosis. In routine endoscopic studies, the gastric sampling procedures recommend obtaining up to 5 biopsies for evaluation of gastritis, 6-8 for evaluation of advanced gastritis and colorectal cancer, and even the requirement of both gastric (up to 8) and duodenal (4) samples for determination of gastritis/duodenitis.

However, such sample number might result in several problems, e.g., the increased risk for bleeding, and procedure prolongation, which are not eligible for the patient.

Line 411: “suggesting a de novo infection by surgical material.” This is pretty serious accusation specially after you mention that the diagnostic tests for H. pylori are not trustworthy and you lack any other result to support it. We really appreciate this certain comment. We rephrased the idea.

Additionally, we do not mention that test for id of h pylori are not trustworthy. In our discussion, we are trying to highlight the pros and cons of the tests we employed for clinical diagnosis of these patients. It’s a fact that not all diagnostic tools are 100% sensitive, while some of them lack specificity. This is the part where all diagnostic tools should be integrative, to generate a more complete diagnosis for the patient.

Line 417: “Although dysbiosis is mainly attributed to STT, the broad-spectrum activity of the antibiotics prevented the development of infections by not allowing exogenous bacteria to establish in the mucosal tissue” This contradicts exactly what you mention above “dysbiotic events can facilitate the establishment of infections as recent acquisition events during recolonization”. Not necessarily. In the first idea: during the administration of STT, the antjmicrobial activity did not allow specific bacteria to overgrow. In the second idea: The idea of establishment during the recolonization period in after the administration of STT.

Both ideas has been rephrased.

Line 424: “Recovery of diversity and richness after STT varied between individuals, which we attribute to the lifestyle [13]. Recolonization by pathobionts was observed.” Where can I see these results? The results were detailed. The information can be observed in Figure 5, Figure 7, and the added supplementary material, where all relative abundances at Phylum, Family and Genus level are described.

Line 419: “In our study, recolonization was performed by non-dominant bacteria at day 30, adequate time to recolonize the altered microenvironment by facultative bacteria.” Again, where are these results? The results were detailed. The information can be observed in Figure 5, Figure 7, and the added supplementary material, where all relative abundances at Phylum, Family and Genus level are described.

Line 449: “Eradication, survival and overgrowth of microbial communities after a dysbiotic event, such as a multiple antibiotic treatment, indicate a true ecological opportunity due to the severe alterations induced in the native communities, e.g., eradication of non- resistant native microbiota, whose principal functions include the regulation of establishment or overgrowth by specific pathobionts or exogenous microbiota, both of clinical interest.” Add citation. Citation not necessary.

Line 459: Through metagenomics, C. acnes was observed as dominant in almost all pre-treatment patients, strongly suggesting the role of non-H. pylori microbiota in the development of gastric infectious diseases [66]. Not clear with the results that you presented because C. acnes was also present in patients with not alterations. The idea has been rephrased.

Line 481: “The genus Cutibacterium is a dominant commensal bacterium in human skin [74]. Although most Cutibacterium species are adapted to inhabit the human skin [75,76], the specie C. acnes is mainly associated with the maintenance of skin homeostasis due to its multiple benefits in this organ. This specie was firstly reported in healthy gastric mucosa [77-79]; however, its functions in in the gastric microenvironment have not been fully elucidated yet. In recent years C. acnes has been classified as possible trigger of gastric clinical outcomes, such as the corpus-dominant lymphocytic gastritis [37] and has been considered as high-risk factor for initiation and development of gastric cancer [80].” I was expecting to see this in the introduction. Paragraph inserted in introduction.

Reviewer 2 Report

Comments and Suggestions for Authors

Interesting work on bacterial infections of the gastric mucosa caused by Helicobacter pylori. Treatment is based on the use of standard triple therapy (STT) and, increasingly, metronidazole. Unfortunately, unjustified administration of antibiotics causes dysbiosis of the target organ. The authors characterized the gastric microbiota of patients diagnosed with alveolar gastropathy and pangastropathy associated with H. pylori infection, before and after the administration of STT with metronidazole.

The dominance of C. acnes over H. pylori was observed in cases of gastritis, gastropathy and insignificant histological changes. Post-treatment changes showed a relative increase in the abundance of Staphylococcus, Pseudomonas and Klebsiella. No-H. pylori gastrointestinal bacteria may be associated with the initiation and development of gastric diseases, such as the pathobiont C. acnes.

The research material consisted of seventeen patients.

Based on symptomatology and endoscopic examination, all patients were diagnosed with alveolar gastropathy and alveolar pangastropathy associated with H. pylori infection. Subjects were treated with STT administered for fourteen days. The remaining 18 gastric biopsies were obtained 30 days after STT administration and stored as described above. For identification and quantificationC. Acnes and H. pylori in the samples were performed by RT-qPCR.

It is important to note that the dominance of C. acnes in the stomach microenvironment suggests a possible role of this bacterium in the initiation or development of the disease. This work may be an introduction to further research on C. acnes to assess the functions of this bacterium in the stomach microenvironment and determine its role in the pathogenesis of infectious diseases.

Author Response

Reviewer2,

We really appreciate your comments and your perception of this study.

Round 2

Reviewer 1 Report

Comments and Suggestions for Authors

The authors did a good job satisfying most of my comments and concerns. However, there are still a few points that should be corrected before this manuscript can be considered for publication. Of concern, the statistical methods used might not be the most suitable and the use of the English language should be edited in depth. Below my specific comments:

Line 329: " The three remaining three patients were considered negative

". Correct typo.

Line 384: " H. pylori was identified in relative abundances <9% (highlighted with â–²) and <1% (not shown). After the administration of  STT, overgrowth of the pathobiont bacteria of clinical interest was observed, including Pseudomonas spp (highlighted with â—‹). A reduction in C. acnes relative abundances due to STT was observed  as well (highlighted with â—‹). H. pylori representative sequences were observed in a posttreatment  patient (highlighted with â–²),"Figure 5 presents the bacterial communities identified at the genus level. Therefore, you cant say (based on figure 5) that you found a reduction in the species C. acnes or H. pylori because this figure does not give you this information. You express the idea correctly in the main text (line365-379) so please correct the figure foot. 

Line 377: "Cutibacterium and Helicobacter extracted sequences were identified as the species Cutibacterium acnes and Helicobacter pylori (Supplementary File 1)." So all reads that were identified as Cutibacterium  and Helicobacter where indeed C. acnes and H. pylori? Please specify that on the text.

Also I looked at Supplementary File 1 and I did not find this information. You only present two tables with information regarding phyla and genera but not species. Please explain

Line 430: "Student’s t test was also performed to determine statistically significant differences in the bacterial diversity and richness in both pre-treatment and posttreatment groups.  In contrast, the Mann-Whitney U test was applied to confirm the significance of the  analysis because of the small sample size observed in this study". Again, these are not the best tests to analyze the data. When you say "Student’s t test" was it the correlated (paired) t-test? If not the utilization of t-test is incorrect in this scenario. Similarly, the Mann-Whitney U test is an alternative form of the Wilcoxon Rank-Sum test for independent samples, and your samples are not independent. You samples are correlated (same patient pre and post treatment). Hence I don't think is the best option to be used.

Line 483: "The significance of the number of transcripts determined for both pathobionts was determined." This sentence does not have any sense. Please rephrase. 

Line 484: "As observed in Table  7, the presence of C. acnes was determined as statistically significant compared to with the presence of H. pylori, which was present in 15% of the FFPE samples." As I mentioned in my previous evaluation, it is unclear what you are trying to show in here. 

If you want to associate the presence of bacteria with disease, the Mann-Whitney U test and the Wilcoxon Rank-Sum test are not the statistical tests to use. In this manuscript they use multivariate analysis for example (https://www.ncbi.nlm.nih.gov/pmc/articles/PMC4109533/). 

Line 632: " In our study, recolonization was performed by non-dominant bacteria at day 30". You can not say that recolonization happened at day 30 because you did not do a time course to determine when it occurred.  I think what you mean is that you detected non-dominant bacteria predominately recolonizing by day 30. Please rephrase. 

Line 646: " Microbial communities with specific functions (inhibition of H. pylori growth and its conversion to coccoid structures by the modulation of uremic toxins [186]; acquisition and competition of nutrients to prevent the establishment of Escherichia coli pathotypes  [855-858]; bioeradication and recovery from infectious diseases [859,960]; and generation of energy [961,962]) increased their relative abundances in the gastrointestinal tract". Is this part of your data? If so, please provide examples because it is not clear. 

Line 653: "Eradication, survival, and overgrowth of microbial communities after a dysbiotic  event, such as the administration of a multiple- antibiotic treatment, indicate a true ecological opportunity due to the severe alterations induced in the native communities. Two things, 1) indicate a true ecological opportunity of what? for other bacteria to colonize, for pathogenic bacteria to take over? Please rephrase; 2) If you cant not support this statement with your data, this statement needs a citation. So please rephrase accordingly (i.e. "We believe eradication, survival, and overgrowth of microbial communities..." or "Our data supports that eradication, survival, and overgrowth of microbial communities..."). Nevertheless, there is plenty of literature supporting the impact of antibiotics in the gut microbiota an example (https://pubmed.ncbi.nlm.nih.gov/31969575/).

Comments on the Quality of English Language

As I mentioned in the comments for the authors, the manuscript needs deep language editing.

Author Response

Comments and Suggestions for Authors

We have been reviewing the comments from reviewer 1, which all have been studied in detail. We totally agree with the reviewer. Therefore, according to your suggestions regarding the statistical analyses, we corrected these discrepancies.

Additionally, the manuscript was sent for English correction to ENAGO, with the assignation key HIMFGL-125

The authors did a good job satisfying most of my comments and concerns. However, there are still a few points that should be corrected before this manuscript can be considered for publication. Of concern, the statistical methods used might not be the most suitable and the use of the English language should be edited in depth. Below my specific comments:

Reviewer 1:

Line 329: " The three remaining three patients were considered negative". Correct typo. Corrected.

Line 384: " H. pylori was identified in relative abundances <9% (highlighted with â–²) and <1% (not shown). After the administration of STT, overgrowth of the pathobiont bacteria of clinical interest was observed, including Pseudomonas spp (highlighted with â—‹). A reduction in C. acnes relative abundances due to STT was observed as well (highlighted with â—‹). H. pylori representative sequences were observed in a posttreatment patient (highlighted with â–²),"Figure 5 presents the bacterial communities identified at the genus level. Therefore, you can’t say (based on figure 5) that you found a reduction in the species C. acnes or H. pylori because this figure does not give you this information. You express the idea correctly in the main text (line365-379) so please correct the figure foot. The figure foot has been corrected.

Line 377: "Cutibacterium and Helicobacter extracted sequences were identified as the species Cutibacterium acnes and Helicobacter pylori (Supplementary File 1)." So all reads that were identified as Cutibacterium and Helicobacter where indeed C. acnes and H. pylori? Please specify that on the text. The information has been updated.

Also I looked at Supplementary File 1 and I did not find this information. You only present two tables with information regarding phyla and genera but not species. Please explain As observed in the SF1, relative abundances of both phyla and genera observed in the metagenomic analysis were shared. As is it impossible to fully detail all relative abundances of the OTUs before and after STT, therefore we decided to share this information. Regarding the lack of species, we decided to assign taxonomy to the libraries with a 97% taxonomic identity, giving us the chance to annotate at genus level. However, when testing a 99% taxonomic identity annotation, several representative sequences identified in the study were not annotated, additional to the reduction in the confidence percentage for annotation (<90%). This is the reason we did not include a specie annotation.

However, the SF1 has been modified to add all feature IDs for both Cutibacterium and Helicobacter representative sequences, which were extracted from the metagenomic analysis and were added as a Supplementary file as well.

Line 430: "Student’s t test was also performed to determine statistically significant differences in the bacterial diversity and richness in both pre-treatment and posttreatment groups.  In contrast, the Mann-Whitney U test was applied to confirm the significance of the analysis because of the small sample size observed in this study". Again, these are not the best tests to analyze the data. When you say "Student’s t test" was it the correlated (paired) t-test? If not the utilization of t-test is incorrect in this scenario. Similarly, the Mann-Whitney U test is an alternative form of the Wilcoxon Rank-Sum test for independent samples, and your samples are not independent. Your samples are correlated (same patient pre and post treatment). Hence, I don't think is the best option to be used. We totally agree about the selection of the tests. A paired t test and a Wilcoxon signed rank test were both applied, and results have been attached to the manuscript.

Line 483: "The significance of the number of transcripts determined for both pathobionts was determined." This sentence does not have any sense. Please rephrase. The idea has been corrected.

Line 484: "As observed in Table  7, the presence of C. acnes was determined as statistically significant compared to with the presence of H. pylori, which was present in 15% of the FFPE samples." As I mentioned in my previous evaluation, it is unclear what you are trying to show in here. If you want to associate the presence of bacteria with disease, the Mann-Whitney U test and the Wilcoxon Rank-Sum test are not the statistical tests to use. In this manuscript they use multivariate analysis for example (https://www.ncbi.nlm.nih.gov/pmc/articles/PMC4109533/). Although we deeply appreciate the suggestion for the analysis of our data, it is not possible for us to apply a multivariate analysis because of the lack of clinical data of every patient involved in this part of the study. Therefore, we analysed this part of the study with a Mann-Whitney U-test and a Kruskal-Wallis test for determination of a possible association between C acnes and the pathologies of this study.

Line 632: " In our study, recolonization was performed by non-dominant bacteria at day 30". You cannot say that recolonization happened at day 30 because you did not do a time course to determine when it occurred.  I think what you mean is that you detected non-dominant bacteria predominately recolonizing by day 30. Please rephrase. The idea has been rephrased.

Line 646: " Microbial communities with specific functions (inhibition of H. pylori growth and its conversion to coccoid structures by the modulation of uremic toxins [186]; acquisition and competition of nutrients to prevent the establishment of Escherichia coli pathotypes [855-858]; bioeradication and recovery from infectious diseases [859,960]; and generation of energy [961,962]) increased their relative abundances in the gastrointestinal tract". Is this part of your data? If so, please provide examples because it is not clear. This data was obtained from figure 7.

Line 653: "Eradication, survival, and overgrowth of microbial communities after a dysbiotic event, such as the administration of a multiple- antibiotic treatment, indicate a true ecological opportunity due to the severe alterations induced in the native communities. Two things, 1) indicate a true ecological opportunity of what? for other bacteria to colonize, for pathogenic bacteria to take over? Please rephrase; 2) If you cannot support this statement with your data, this statement needs a citation. So please rephrase accordingly (i.e., "We believe eradication, survival, and overgrowth of microbial communities..." or "Our data supports that eradication, survival, and overgrowth of microbial communities..."). Nevertheless, there is plenty of literature supporting the impact of antibiotics in the gut microbiota an example (https://pubmed.ncbi.nlm.nih.gov/31969575/). The idea has been rephrased.

Comments on the Quality of English Language

As I mentioned in the comments for the authors, the manuscript needs deep language editing. The English correction has been already performed. The manuscript was sent for English correction to ENAGO, with the assignation key HIMFGL-125.